# A global monthly 3D-field of seawater pH over 3 decades: a machine learning approach

Guorong Zhong[1,2,3,4], Xuegang Li[1,2,3,4*], Jinming Song[1,2,3,4*], Baoxiao Qu[1,2,3,4], Fan Wang[1,2,3,4], Yanjun Wang[1,4], Bin Zhang[1,4], Lijing Cheng[4,5], Jun Ma[1,2,3,4], Huamao Yuan[1,2,3,4], Liqin Duan[1,2,3,4], Ning Li[1,2,3,4], Qidong Wang[1,2,3,4], Jianwei Xing[1,2,3,4], Jiajia Dai[1,2,3,4]

[1]Institute of Oceanology, Chinese Academy of Sciences, Qingdao 266071, China
[2]Laboratory for Marine Ecology and Environmental Science, Qingdao National Laboratory for Marine Science and Technology, Qingdao, China
[3]University of Chinese Academy of Sciences, Beijing 101407, China
[4]Center for Ocean Mega-Science, Chinese Academy of Sciences, Qingdao 266071, China
[5]Institute of Atmospheric Physics, Chinese Academy of Sciences, Beijing 100029, China

*Correspondence to: Xuegang Li (lixuegang@qdio.ac.cn) and Jinming Song (jmsong@qdio.ac.cn)*

**Abstract.** The continuous uptake of anthropogenic $CO_2$ by the ocean leads to ocean acidification, which is an ongoing threat to the marine ecosystem. The ocean acidification rate was globally documented in the surface ocean but limited below the surface. Here, we present a monthly four-dimensional $1°×1°$ gridded product of global seawater pH at total scale and in-situ temperature (without standardization to 25°C), derived from a machine learning algorithm trained on pH observations from the Global Ocean Data Analysis Project (GLODAP). The proposed pH product covers the years 1992-2020 and depths from the surface to 2 km on 41 levels. A three-step machine learning-based algorithm was used to construct the pH product, incorporating region division by the self-organizing map neural network, predictor selection by the stepwise regression algorithm that adds and removes variables from network inputs based on their contribution to reducing reconstruction errors, and non-linear relationship regression by feed-forward neural networks (FFNN). The performance of the machine learning algorithm was validated using real observations by a cross validation method, where four repeating iterations were carried out with 25% varied observations for each evaluation and 75% for training. The proposed pH product is evaluated through comparisons to time series observations and the GLODAP pH climatology. The overall root mean square error between the FFNN reconstructed pH and the GLODAP measurements is 0.028, ranging from 0.044 in the surface to 0.013 at 2000 m. The pH product is distributed through the data repository of the Marine Science Data Center of the Chinese Academy of Sciences at http://dx.doi.org/10.12157/IOCAS.20230720.001 (Zhong et al., 2023).

## 1 Introduction

Since the Industrial Revolution, the oceans have absorbed approximately one-quarter of the carbon dioxide emitted by human activities (Le Quéré et al., 2010; Friedlingstein et al., 2023). The continuous absorption of carbon dioxide from the atmosphere results in a decline in carbonate saturation states and surface seawater pH, which is a phenomenon of great concern: ocean acidification (Caldeira et al., 2003; Feely et al., 2004; Orr et al., 2005; Feely et al., 2009). As one of the primary environmental challenges the ocean faces today, ocean acidification will have extensive impacts on marine organisms and the ecological environment, resulting in notable changes to the marine ecosystem. Therefore, the assessment of ocean acidification is crucial for researching the response of marine organisms to changes in seawater pH and understanding the potential future changes in the capacity of the global ocean to uptake $CO_2$ (Sabine et al., 2010; Guallart et al., 2015).

However, acidification research is greatly limited in terms of temporal and spatial coverage due to the lack of long-term, global coverage, and continuous seawater pH measurements. Accurate seawater pH measurements are only available from select ship surveys and a limited number of time series stations in recent decades (Fay et al., 2013; Takahashi et al., 2014). Recent research using discrete ship survey measurements revealed rapid surface ocean acidification in the Arctic Ocean, with

some areas showing an average decreasing pH trend of -0.0086 yr$^{-1}$ (Luo et al., 2016; Terhaar et al., 2020; Qi et al., 2022). Both seawater pH measurements from time series stations and discrete ship surveys suggest notable regional differences in surface ocean acidification rates (Bates et al., 2014; Lauvset et al., 2015). In the Japan/East Sea, the acidification rate in the deep ocean may be faster than previously considered and even faster than in the surface ocean (Chen et al., 2017; Li et al., 2022). Meanwhile, relatively slow acidification was found in the deep Atlantic Ocean below 2000 m (Guallart et al., 2015), and rising pH in deep waters around 1000 m was also reported in the North Pacific Ocean (Ishizu et al., 2021). With limited reports about acidification below the surface, there remains a need to enhance our understanding of global ocean acidification rates across varying depths.

The lack of long-term, global coverage, and continuous seawater pH measurements makes it difficult to expand the understanding of global deep ocean acidification using classic regression methods. Recent applications of machine learning methods in global reconstructions of marine carbonate system variables have facilitated global-scale research on the acidification and carbon cycle, including the single/ensemble-based FFNN method and the SOM-FFNN method for reconstruction of surface ocean partial pressure of $CO_2$ ($p$CO$_2$, Landschützer et al., 2014; Chau et al., 2022; Zhong et al., 2022; Chau et al., 2024), dissolved inorganic carbon (DIC, Broullón et al., 2020; Keppler et al., 2020; Gregor and Gruber, 2021; Chau et al., 2024), and alkalinity (Broullón et al., 2019; Gregor and Gruber, 2021; Chau et al., 2024). These methods have inspired our methodology for constructing the global gridded seawater pH dataset. Until now, only surface ocean gridded pH products are available in acidification research, including the 1° JMA product (Iida et al., 2021), the 1° OceanSODA-ETHZ product (Gregor and Gruber, 2021), the 0.25° remote-sensing-based product (Jiang et al., 2022), and the 0.25° CMEMS-LSCE product (Chau et al., 2024), which were derived from reconstructing $p$CO$_2$, DIC, or alkalinity using machine learning algorithms and subsequently calculating pH with the CO2SYS program (Lewis and Wallace, 1998). In this paper, we present a monthly gridded global ocean pH product covering depths of 0-2000 m from January 1992 to December 2020, using a machine learning method trained on pH measurements from the Global Ocean Data Analysis Project (GLODAP) dataset (Lauvset et al., 2023). The proposed pH product provides regional and global insight into ocean acidification on timescales ranging from a few years to multiple decades.

## 2 Methods

### 2.1 Data sources and processing

The pH measurements at total scale and in-situ temperature and pressure from the Global Ocean Data Analysis Project (GLODAP) dataset 2023 version were used for neural network training (Lauvset et al., 2023). The reconstructed pH product is also at total scale and in-situ temperature (without standardization to 25°C) based on a gridded global seawater temperature product (Cheng et al., 2017). We have collected gridded products of different variables as potential pH predictors (Table 1), and the selection of these products was based on two reasons. The first reason was their potential association with physical, chemical, and biological ocean process which may affect the seawater pH. Another reason was the sufficient availability in time and spatial coverage and their potential association with the unavailable interannual variability of some climatological products used. Specifically, the mixed layer depth, bathymetry, and ocean currents were related to the physical mixing of seawater and spatial distribution of pH. Sea level pressure, surface pressure, wind speed, sea surface height, surface ocean $p$CO$_2$, and dry air mixing ratio of atmospheric $CO_2$ were related to the $CO_2$ exchange across the interface. The Multivariate ENSO index, Arctic Oscillation index, and Southern Oscillation index may be related to pH variability over years or decades in particular regions. The total alkalinity and DIC reflect the ocean carbonate system and were generally used to calculate seawater pH indirectly. However, 3D field products with sufficient time and spatial coverage are currently not available for these two variables, so monthly climatological 3D products were used for better pH spatial distribution. The remote sensing products are related to biological production of organic matter, including chlorophyll concentration, diffuse attenuation coefficient, remote sensing reflectance, and total absorption/ backscattering. Temporal and spatial sample information,

including latitude, longitude, depth and sample time, was also used as supplementary variables. Latitude and longitude were normalized to radians using sine and cosine transformations, to present connected sample position information. The spatial sample position and time information of GLODAP measurements were input in the training of FFNNs, and the spatial position and time of defined 1° and monthly product grids were input into FFNNs during the interpolation process to output a gridded product. Most predictor products were obtained with a monthly and 1°×1°resolution, which can be directly used without any treatments. Differently, products with higher resolutions were integrated into the same monthly and 1°×1° resolution by averaging, before they can be used in the relationship fitting. For instance, the mixed layer depth product, originally obtained with a resolution of 0.25°×0.25°, was converted to a 1°×1° resolution by averaging 16 0.25° grids into one 1° grid. Similarly, such as the $xCO_2$ product, predictor products obtained with weekly resolutions were converted to the monthly resolution by directly averaging all values within the same month. Products used for variables listed in Table 1 was chose due to their sufficient temporal and spatial coverage and the application in previous research on reconstruct of carbonate system variables. For example, the ECCO2 MLD product has been used in reconstructions of the CMEMS-LSCE surface ocean carbonate system variables product (Chau, et al., 2024) and the MPI-SOM-FFN $pCO_2$ product (Landschützer et al., 2014).

**Table 1. Data products used as pH predictors.**

| Predictor | Abbreviation | Data product and reference | Resolution | Related process to affect pH |
|---|---|---|---|---|
| Sine of (latitude · π/180°) | sin(Lat) | - | - | Sample position and time of GLODAP pH measurements |
| Sine of (longitude · π/180°) | sin(Lon) | - | - | |
| Cosine of (longitude · π/180°) | cos(Lon) | - | - | |
| Number of months since January 1992 | $N_{mon}$ | - | - | |
| Year | Year | - | - | |
| Month | Month | - | - | |
| Depth | Depth | - | - | |
| Temperature and monthly anomaly | Temp, $Temp_{anom}$ | IAP global ocean temperature gridded product (Cheng et al., 2016; 2017) | 1°, monthly since 1940, 0-2000 m with 41 levels | State of carbonate system |
| Salinity and monthly anomaly | Sal, $Sal_{anom}$ | IAP global ocean salinity gridded product (Cheng et al., 2020) | 1°, monthly since 1940, 0-2000 m with 41 levels | |
| Climatological total alkalinity | Alk | AT_NNGv2_climatology (Broullón et al., 2019) | 1°, monthly climatological, 0-5500 m with 102 levels | |
| Climatological dissolved inorganic carbon | DIC | TCO2_NNGv2LDEO_climatology (Broullón et al., 2020) | 1°, monthly climatological, 0-5500 m with 102 levels | |
| Climatological dissolved oxygen | DO | WOA18 (Garcia et al., 2020a) | 1°, monthly climatological, 0-5500 m with 102 levels | Biological production and drawdown of organic matter |
| Climatological nitrate | Nitrate | WOA18 (Garcia et al., 2020b) | 1°, monthly climatological, 0-5500 m with 102 levels | |
| Climatological phosphate | Phosphate | | | |
| Climatological silicate | Silicate | | | |
| Mixed layer depth and monthly anomaly | MLD, $MLD_{anom}$ | ECCO2 cube92 (Menemenlis et al., 2008) | 0.25°, monthly since 1992 | Physical mixing of seawater and stratification |
| Sea surface height and monthly anomaly | SSH, $SSH_{anom}$ | | | ocean wave, tides, current, and sea-level rise |
| W velocity of ocean currents at 5 m, 65m, 105m, 195m, and in-situ depth | $W_{vel}(5m)−W_{vel}(in\text{-}situ)$ | | | Ocean current and upwelling |
| Sea level pressure | SLP | ERA5 (Hersbach et al., 2020) | 1°, monthly since 1940 | $CO_2$ exchange between surface seawater and atmosphere |
| Surface pressure | $P_{surf}$ | | | |
| dry air mixing ratio of atmospheric $CO_2$ and monthly anomaly | $xCO_2$, $xCO_{2\ anom}$ | NOAA Greenhouse Gas Marine Boundary Layer Reference (Lan et al., 2023) | 0.25°, weekly since 1979 | |
| Multivariate ENSO Index | MEI | bi-monthly Multivariate El Niño/Southern Oscillation index (Wolter et al., 2011) | monthly since 1979 | El Niño and Southern Oscillation |
| Arctic Oscillation index | AOI | Climate Prediction Center Daily Arctic Oscillation Index (CPC, 2002) | monthly since 1950 | Arctic Oscillation |

| Southern Oscillation Index | SOI | Climate Prediction Center Southern Oscillation Index (CPC, 2005) | monthly since 1951 | Southern Oscillation |
|---|---|---|---|---|
| Bathymetry | Bathy | GEBCO_2022 Grid (GEBCO, 2022) | 15 arc-second | Vertical volume of seawater |
| 10 m Wind speed and monthly anomaly | Wind, Wind$_{anom}$ | ERA5 (Hersbach et al., 2020) | 1°, monthly since 1940 | $CO_2$ exchange between surface seawater and atmosphere |
| Surface ocean $pCO_2$ | $pCO_2$ | Stepwise FFNN (Zhong et al., 2022) | 1°, monthly since 1992 | |
| Climatology of Surface Ocean $pCO_2$ | $pCO_2$ clim | MPI-ULB-SOM_FFN_clim (Landschützer et al., 2020) | 0.25°, monthly climatological | |
| Chlorophyll and monthly anomaly[*] | Chl, Chl $_{anom}$ | MODIS-Aqua Chlorophyll Data (NASA, 2022a) | 9km, monthly since 2002 | Biological production of organic matter |
| Photosynthetically Available Radiation | PAR | MODIS-Aqua Photosynthetically Available Radiation Data (NASA, 2022b) | | Light penetration and availability in aquatic systems influencing phytoplankton photosynthesis |
| Diffuse attenuation coefficient at 490 nm | KD490 | MODIS-Aqua Downwelling Diffuse Attenuation Coefficient Data (NASA, 2022c) | | |
| Remote sensing reflectance at 412-678 nm[**] | RRS412−RRS678 | MODIS-Aqua Remote-Sensing Reflectance Data (NASA, 2022d) | | Phytoplankton composition and suspended particulate matter, indicators of biological productivity |
| Total absorption at 412-678 nm | Ta412−Ta678 | MODIS-Aqua Inherent Optical Properties Data (NASA, 2022e) | | |
| Total backscattering at 412-678 nm | Tb412−Tb678 | MODIS-Aqua Inherent Optical Properties Data (NASA, 2022e) | | |

[*]: products from Chlorophyll to Total backscattering are satellite remote sensing products;

[**]: Remote sensing reflectance, total absorption, and total backscattering both include 10 wavelengths: 412nm, 443nm, 469nm, 488nm, 531nm, 547nm, 555nm, 645nm, 667nm, and 678nm, with each wave length regard as one individual parameter.

On the other hand, the discrete GLODAP measurements did not match the monthly 1°×1° resolution of pH predictor products. To be consistent in the temporal and spatial resolution, the discrete GLODAP measurements were also merged into a monthly and 1°×1° resolution by averaging. The vertical layer of the temperature and salinity gridded product were used as reference standards for adjusting other collected products and constructing the pH product (Cheng et al., 2016; Cheng et al., 2017; Cheng et al., 2020). These layers covered a depth range of 0-2000 m depth, with a total of 41 layers, including 0 m, 5 m, 10-100 m at 10 m intervals, 120-200 m at 20 m intervals, 250-900 m at 50 m intervals, and 1000-2000 m at 100 m intervals. Subsequently, the in-situ seawater measurements of pH, temperature, salinity, latitude, longitude, and depth from the GLODAP dataset were averaged monthly within the same 1°×1° grid (first grid centered at 89.5°S, 0.5°E) and within the same vertical layer to match the resolution of the predictor products. Since a direct average was used instead of a weighted average, the average latitude, longitude, and depth values from the initial measurements within the same 1°×1° grid were then used as new sample position for the derived monthly measurements, instead of being located at the center point of grids. The pH measurements obtained after the 1°×1° grid and monthly averaging were employed to establish a neural network model and fit a non-linear relationship with the pH predictors.

## 2.2 Biogeochemical province

To identify predictors that are most relevant to pH drivers in different regions, we divide the global ocean into distinct biogeochemical provinces using self-organizing map neural networks (SOM). This was achieved by inputting climatological surface seawater temperature, salinity, mixed layer depth, chlorophyll concentration, dissolved oxygen, nitrate, phosphate, silicate, and pH (Lauvset et al., 2016) into a 4×4 SOM network, resulting in the partitioning of the global ocean into preliminary

16 provinces. Subsequently, the small "island" provinces with fewer than ten connected grids or covered by fewer than 100 GLODAP pH measurements were merged with the nearest neighboring provinces, as the pH reconstruction errors tend to be notably higher due to the extremely few training samples in the non-linear relationship fitting by networks. In addition, the province separated by continents was manually subdivided into distinct provinces, such as the province spanning the North

Pacific and the North Atlantic. As a result, the global ocean was divided into 14 biogeochemical provinces, as shown in Figure 1. The boundary of SOM provinces was treated with a cross-boundary method to relieve the discontinuity of spatial distribution near the SOM boundaries (Zhong et al., 2022). Due to much more dynamic variation in coastal seawater pH, the global coastal areas have higher reconstruction errors than the open oceans. In this study, we removed all coastal areas shallower than 200 m bathymetry. Furthermore, because the drivers of seawater pH near the surface is different with deeper waters, the ocean area

was divided into two layers: the mixed layer (ranging from 0 m to mixed layer depth) and the intermediate layer (ranging from mixed layer depth to 2000 m). Consequently, the gridded product construction in each province was carried out separately for the two layers. Application of SOM method can effectively reduce regional reconstruction errors, but it also generates discontinuity problems near the boundary. Therefore, a cross-boundary method was used to improve the FFNN performance near the SOM and vertical boundary (Zhong et al., 2022). The spatial scale of training samples in each SOM province was

expand out of the boundary for 10 grids, and out of the vertical boundary for 2 layers (Figure 2). By increasing additional training sample outside the SOM province and vertical layer boundary, the cross-boundary method can effectively reduce the appearance of dysconnectivity near boundaries (Figures S1 and S2).

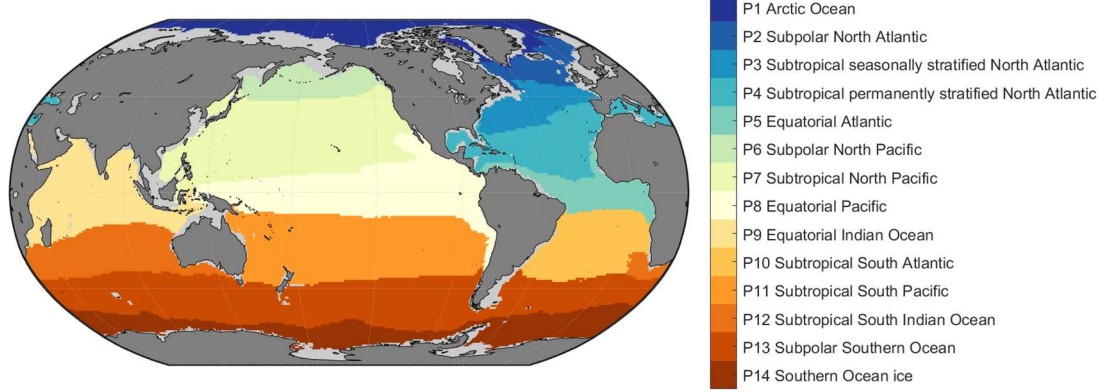

Figure 1: Map of the biogeochemical province.

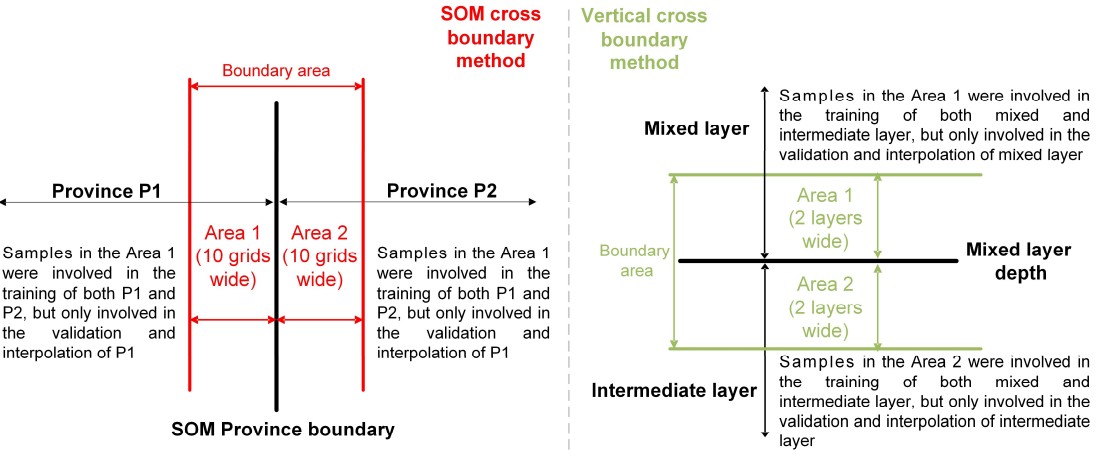

Figure 2: Cross-boundary method for better connectivity near the SOM boundary and vertical boundary.

## 2.3 pH product construction

The forward feedback neural network (FFNN) with a single hidden layer was applied to fit the non-linear relationship between seawater pH and its predictors to perform spatial interpolation and construct the gridded product:

$$pH = f(Predictors_1, Predictors_2, \dots, Predictors_N) \tag{1}$$

where $f$ was a non-linear function built by FFNN, and predictors related to chemical, physical, and biological properties were selected from products in Table 1. Considering the regional difference in pH variability and its drivers, identifying the combination of most relevant predictors in each region was a critical precondition. Thus, the entire product construction method includes two steps (Figure 3):

(1) Selection of seawater pH predictors in each province using the Stepwise FFNN algorithm (referred as (1) Stepwise FFNN in Figure 3). All the collected products were input into the Stepwise FFNN algorithm to identify the predictors that yield the lowest reconstruction errors for seawater pH (Zhong et al., 2022). The variation in standard deviation (MAE) calculated by the K-fold cross validation method will feed back to update the input products. The input variables are selected as pH predictors one by one in the way MAE decreases the fastest. Specifically, by comparing reconstruction errors of using each collected environmental variable in Table 1 as the only predictor input to the FFNN, the variable with the lowest error is selected as the first pH predictor and moved out from the environmental variables list used in the subsequent steps. Subsequently, while keeping the first predictor unchanged, compare reconstruction errors when using each remaining environmental variable as the second input for the FFNN. The variable with the lowest error is determined to be the second pH predictor. In the same way, new predictors are sequentially determined. This selection process continued through multiple iterations until no further reduction in MAE was observed, regardless of whether a variable was added or removed. The variables identified in previous iterations were then output as the optimal pH predictors. Since both overfitting caused by co-correlation and underfitting caused by an insufficient number of predictors result in significant increases in pH reconstruction errors, the lowest reconstruction error is considered to occur between these two states. In order to eliminate potential co-correlation and prevent overfitting, whenever after a new predictor is identified, the algorithm also tests whether the reconstruction error will decrease when sequentially removing each determined predictor. The algorithm individually removes each previously identified predictors immediately after adding one variable as a predictor. If the error decreases after removing a previously determined predictor, this predictor is highly correlated with other identified predictors. If a certain predictor is highly correlated with existing predictors, this predictor tends to fail to compete with other variables in the adding of predictors and is generally removed in the following removal step to reduce reconstruction errors. Therefore, most of the co-correlation among the selected predictors has been removed in this Stepwise FFNN selection procedure. If products with co-correlations are still selected, some products may provide important additional information in specific regions, leading to a greater reduction in reconstruction errors compared to the increase caused by overfitting. Spatial and temporal variables, such as latitude, longitude, and time, are directly related to the spatial or temporal pH patterns rather than the factor driving pH variations. This means these variables are often co-correlated with other input environmental variables. In some regions where the environmental variables sufficiently reflect the factors influencing pH or where spatial and temporal pH patterns are not notable, adding latitude, longitude, and time as predictors does not contribute sufficient information and cannot effectively reduce predicting errors due to the co-correlation with other predictors. In this case, these spatial-temporal variables are not selected as predictors (Tables 2 and 3). In addition, depth is important in reconstructing the vertical pH distribution. However, it was not used as a predictor in certain regions of the mixed layer due to the notable similarity between the vertical pattern of pH and particular environmental variables used as predictors, such as phosphate, nitrate, and silicate. In this case, the FFNN model learned how pH varied with depth based on the similarity of vertical pattern between seawater pH and specific physical or biological conditions indicated by input environmental variables, and subsequently reconstructed seawater pH values at different depths using 3D fields of these environmental variables. In each province, pH predictors were selected separately for the mixed layer (Table 2) and intermediate layer (Table 3). In certain polar areas and prior to August 2002 when satellite remote sensing products (products from $Z_{eu}$ to Tb678 in Table 1) were not available, the additional selection of predictors was carried out without the use of satellite remote sensing products (Table S1). These satellite products were not used in the intermediate layer due to low correlation with seawater pH, with no need for additional selection.

(2) Fitting the non-linear relationship between seawater pH and selected predictors (referred as (2) FFNN in Figure 3). In each province, a group of FFNNs were trained separately for the mixed layer and intermediate layer to fit the non-linear relationship, based on the predictors selected in the first step and GLODAP pH measurements. To mitigate the influence of the FFNN's initial state on reconstructed values, multiple networks with the same structure but different initial states were trained and their results were averaged (Standard deviation showing in Figure S5). Subsequently, the seawater pH was calculated by inputting the product of pH predictors into the trained FFNNs. Since the satellite remote sensing products used in this work lack data during the period before August 2002 and in certain polar areas during winter, the FFNN generated missing values in these grids when remote sensing products were used as predictors. To address these missing values, we selected additional groups of predictors after removing remote sensing products (Table S1), and then trained additional FFNNs to predict pH in grids with missing values. This procedure was the same as the reconstruction process in the intermediate layer, in which the remote sensing products were also not used. Finally, the seawater pH values from all FFNNs were combined to construct the global ocean 0-2000 m seawater pH gridded product from January 1992 to December 2020, with a 1°×1° spatial resolution. The pH data earlier than 1992 is unavailable because the predictors used from ECCO2 cube92 product (Menemenlis et al., 2008) also start from 1992. Data after 2020 is limited by the coverage of used surface ocean $p$CO$_2$ product and will be updated in future works.

All FFNNs used in these two steps have the same structure with a single hidden layer, as using deeper structures tends to cause overfitting and increase pH reconstruction errors. The number of neurons was determined by comparing reconstruction errors of FFNNs with different neurons based on the same training samples, testing samples, and pH predictors, and then adopting the number with the lowest reconstruction error. Specifically, for the stepwise FFNN regression step, the number of neurons in FFNNs was determined using provisional predictors from preliminary experiments with the number of neurons set to 25.

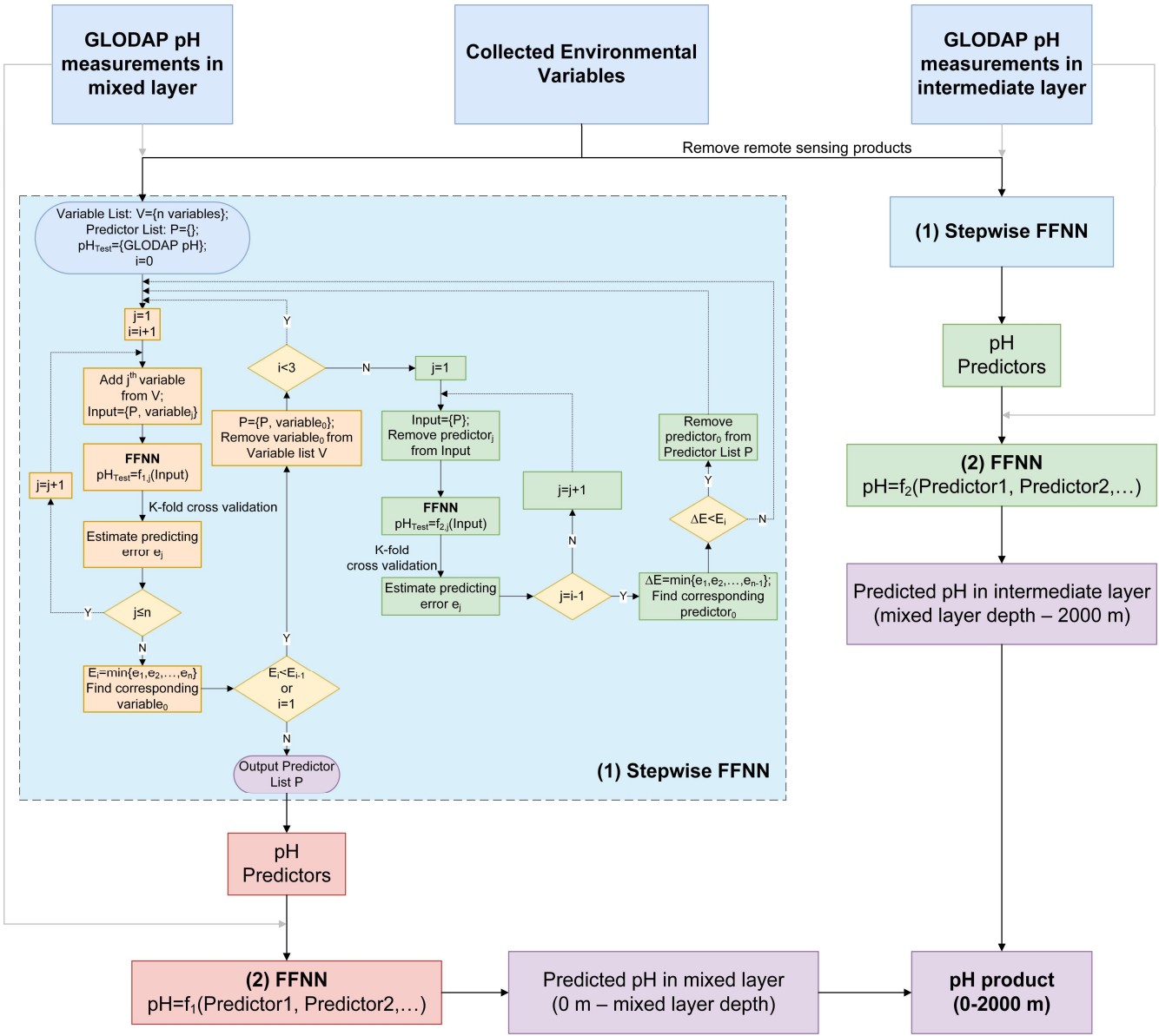

Figure 3: The procedure of pH product construction. (1) Stepwise FFNN: the algorithm for selecting predictors (Zhong et al., 2022); (2) FFNN: fitting the non-linear relationship between seawater pH and its predictors. Collected Environmental variables: collected products listed in Table 1. pH predictors: the selected most informative variables listed in Tables 2 and 3. Remote sensing products: variables from Chlorophyll to Total backscattering in Table 1. Mixed layer: from 0 m to mixed layer depth; intermediate layer: from mixed layer depth to 2000 m.

**Table 2. Predictors selected by the stepwise FFNN algorithm in the Mixed layer.**

| Province | FFNN neurons | pH Predictor |
|---|---|---|
| P1 Arctic Ocean | 10 | $p$CO$_2$, sin(Lat), Depth, Sal, W$_{vel}$(105m) |
| P2 Subpolar North Atlantic | 10 | Phosphate, DO, N$_{mon}$, DIC, Sal, Bathy |
| P3 Seasonally stratified North Atlantic | 75 | $p$CO$_2$ $_{clim}$, Depth, Temp, Silicate, $p$CO$_2$, DIC |
| P4 Permanently stratified North Atlantic | 20 | $p$CO$_2$, Phosphate, sin(Lat), Depth, SSH$_{anom}$, Sal$_{anom}$, W$_{vel}$(195m), Temp, W$_{vel}$(in-situ), $p$CO$_2$ $_{clim}$, DO |
| P5 Equatorial Atlantic | 50 | sin(Lat), Tb469, Temp, Tb555, Tb547, Nitrate, Tb667, Tb678, Tb488, Tb645, Tb531, Sal |
| P6 Subpolar North Pacific | 10 | DIC, sin(Lat), sin(Lon), Depth, Salinity, Temp, $p$CO$_2$, W$_{vel}$(in-situ) |
| P7 Subtropical North Pacific | 50 | Temp, sin(Lon), sin(Lat), $p$CO$_2$, Phosphate, Sal, $p$CO$_2$ $_{clim}$, Depth, cos(Lon), Nitrate, Sal$_{anom}$, Alk |
| P8 Equatorial Pacific | 20 | $p$CO$_2$, Silicate, Depth, Sal, Temp, Wind, Alk, RRS645, Ta555, Ta547 |
| P9 Equatorial Indian Ocean | 10 | DO, Temp$_{anom}$, $p$CO$_2$, Depth, W$_{vel}$(in-situ), W$_{vel}$(195m), W$_{vel}$(65m) |

| | | |
|---|---|---|
| P10 Subtropical South Atlantic | 10 | $p$CO$_2$, DIC, Silicate, RRS645, W$_{vel}$(in-situ), Ta547, Temp, Ta667, Sal, Phosphate, Tb412, Ta412, Tb443, DO, xCO$_2$ |
| P11 Subtropical South Pacific | 10 | Silicate, $p$CO$_2$, Tb412, Phosphate, Depth, Ta488, Temp$_{anom}$, Ta531 |
| P12 Subtropical South Indian Ocean | 10 | $p$CO$_2$, Silicate, Phosphate, Nitrate, Depth, Wind |
| P13 Subpolar Southern Ocean | 20 | Phosphate, Depth, $p$CO$_2$, $p$CO$_2$ clim, Sal, DIC, Nitrate |
| P14 Southern Ocean ice | 20 | Phosphate, Temp, $p$CO$_2$, Depth, Sal, Alk, SSH |

(The predictors are arranged in order of relative importance, with the variables listed at the front of each province being more effective in reducing reconstruction errors when used as pH predictors.)

**Table 3. Predictors selected by the stepwise FFNN algorithm in the intermediate layer.**

| Province | FFNN neurons | pH Predictor |
|---|---|---|
| P1 Arctic Ocean | 50 | Phosphate, Nitrate, Sal, Depth, sin(Lat), SSH |
| P2 Subpolar North Atlantic | 20 | Phosphate, DO, Depth, Year, Sal, Temp, Nitrate, sin(Lat), Alk, W$_{vel}$(195m) |
| P3 Seasonally stratified North Atlantic | 10 | DIC, Nitrate, Temp, Depth, sin(Lon), Year |
| P4 Permanently stratified North Atlantic | 20 | Phosphate, Temp, Depth, sin(Lat), N$_{mon}$, sin(Lon), Sal, Sal$_{anom}$, Nitrate, W$_{vel}$(in-situ) |
| P5 Equatorial Atlantic | 25 | Depth, DIC, Sal, sin(Lat), Temp, Phosphate, SSH, cos(Lon), Nitrate, Silicate |
| P6 Subpolar North Pacific | 25 | Phosphate, Sal, Depth, Temp, sin(Lat), Silicate, xCO$_2$ anom, Alk, Nitrate |
| P7 Subtropical North Pacific | 50 | Phosphate, Sal, Temp, Silicate, N$_{mon}$, sin(Lat), sin(Lon), Depth, Alk, DIC, Nitrate |
| P8 Equatorial Pacific | 25 | Phosphate, Depth, Temp, sin(Lat), Sal, Silicate, xCO$_2$, Nitrate, W$_{vel}$(105m) |
| P9 Equatorial Indian Ocean | 10 | Phosphate, Depth, $p$CO$_2$, W$_{vel}$(in situ) |
| P10 Subtropical South Atlantic | 10 | Temp, DIC, Sal, Depth, Nitrate, W$_{vel}$(65m), $p$CO$_2$, $p$CO$_2$ clim, DO, W$_{vel}$(195m) |
| P11 Subtropical South Pacific | 25 | Phosphate, Depth, Temp, xCO$_2$, sin(Lat), Silicate, Sal, Alk |
| P12 Subtropical South Indian Ocean | 25 | Phosphate, $p$CO$_2$, Depth, Temp, Sal, $p$CO$_2$ clim, Silicate, DO |
| P13 Subpolar Southern Ocean | 50 | DIC, Temp, Depth, N$_{mon}$, Sal, Alk, DO, Silicate, P$_{surf}$, Temp$_{anom}$ |
| P14 Southern Ocean ice | 25 | cos(Lon), sin(Lat), Depth, DIC, Temp, Sal |

(The predictors are arranged in order of relative importance, with the variables listed at the front of each province being more effective in reducing reconstruction errors when used as pH predictors.)

**2.4 Validation and uncertainty**

The reconstructed pH product was validated based on pH measurements from GLODAP and time series stations. First, the root mean square error (RMSE) between the FFNN pH and GLODAP pH measurements was calculated using the K-fold cross validation method. The GLODAP pH measurements were divided by years, and the K value was 4 to keep aside 25% independent measurements for testing in each one of the total 4 iterations. Thus, within every set of four consecutive years, pH measurements from three years were utilized for training the FFNN model, while the measurements from the remaining year were employed for testing. This approach ensured the independence between the training and testing groups (Gregor et al., 2019; Zhong et al., 2022). Subsequently, the pH measurements in the testing group were compared against the FFNN pH values based on the training group. A total of 4 iterations were carried out with each iteration designating different years as the testing groups, ensuring that measurements from all years have been set as the test group once and matched with a FFNN value. By comparing all FFNN pH values with GLODAP pH measurements, the RMSE of pH and the molar hydrogen ion concentration ([H$^+$]) was calculated to evaluate the performance of the FFNN model. The reconstruction of the testing group from the training group is similar to the interpolation process, wherein the FFNN is trained with existing measurements to reconstruct pH in unknown areas.

Second, the reconstructed seawater pH product was compared with independent pH measurements from the Hawaii Ocean Time-series (HOT, 22° 45' N, 158° 00' W, since October 1988) (Dore et al., 2009), Bermuda Atlantic Time-series Study (BAT, 31°50' N, 64°10' W, since October 1988) (Bates et al., 2007; Bates et al., 2020), and The European Station for Time Series in the Ocean Canary Islands (ESTOC, 29°10' N, 15°30' W, from 1995 to 2009) (González-Dávila et al., 2010). The long-term trend was further compared with data from the Irminger Sea station (64.3°N, 28.0°W, from 1983 to 2019, Ólafsson, 2016; Ólafsdóttir et al., 2020a), the Iceland Sea station (68.0°N, 12.7°W, from 1985 to 2019, Ólafsson, 2012; Ólafsdóttir et al., 2020b), and the DYFAMED station (42.3°N, 7.5° E, from 1991 to 2017, Coppola et al., 2024). For better evaluating the performance of FFNN below the surface, the constructed pH product was also compared to independent delayed-mode pH-adjusted data with quality control flag 1 from the biogeochemical-Argo (BGC-Argo) profiles from Global Data Assembly Centre (Claustre et al., 2020; Argo, 2024). Validation based on these independent measurements from time series stations and BGC-Argo profiles provides additional evidence of data accuracy.

A comparison between the method of training FFNN with pH and the method of training FFNN with $[H^+]$ then converting to pH was carried out, to validate which way has a lower pH reconstruction error (Figure S3). In addition, to identify the difference in pH variability uncertainty hidden by logarithm among regions with the same pH RMSE but different pH level, the uncertainty of reconstructed pH values was converted from $[H^+]$ RMSE instead of directly using pH RMSE. The pH obtained from the FFNN was first converted to $[H^+]$ to estimate RMSE. Subsequently, the pH values were shown as $pH_0 \pm \sigma$ at each given $pH_0$ value, and the local uncertainty $\sigma$ stem from FFNN reconstruction errors was calculated as the following:

$$\sigma = -\log_{10}(10^{-pH_0} - RMSE_{[H^+]}) - pH_0 \qquad (2)$$

where $RMSE_{[H^+]}$ was the RMSE of $[H^+]$ converted from FFNN pH in each layer of all 14 biogeochemical provinces, $pH_0$ was the local FFNN predicted pH value. The local uncertainty $\sigma$ calculated by this method is simultaneously related to the pH reconstruction error and local pH level which serves to convert the overall province FFNN error into local errors and better distinguishes the differences in uncertainty across different regions. The uncertainty of products used as pH predictors is one ineluctable source for pH reconstruction errors of the FFNN model. However, the direct estimation of pH uncertainty from summing the uncertainty of each used product is not feasible. Combining the inherent uncertainties of different predictor products via error propagation relies on the partial derivatives of pH to each predictor, but the non-linear relationships established by the FFNN do not have a specific formula, leading to the difficulty in calculating the partial derivatives. Therefore, the local uncertainty of our pH product was directly estimated from the regional FFNN pH reconstruction errors and local pH values following formula (2), instead of synthesizing the inherent uncertainty of each used predictor product through the propagation of errors. The inherent uncertainty and construction method of predictor products are described in the Supplementary text.

## 3 Results and discussion
### 3.1 Validation of algorithm
### 3.1.1 Validation based on GLODAP and time series measurements

Compared with the GLODAP dataset, most reconstructed values of Stepwise FFNN are close to the GLODAP pH measurements, concentrated around the y=x line (Figure 4). Only a few samples notably differ between the pH measurements and the reconstructed values, with the RMSE of 0.028 in the global ocean of 0-2000 m. A better performance of the FFNN was found in the intermediate layer, with testing samples more concentrated on the y=x line. The RMSE in the mixed layer is 0.034, higher than 0.026 in the intermediate layer. The minor difference between the reconstructed value and the pH measurements and the $R^2$ of 0.97 in the intermediate layer may be caused by less pH variability at depth and better model fit with broader pH value range.

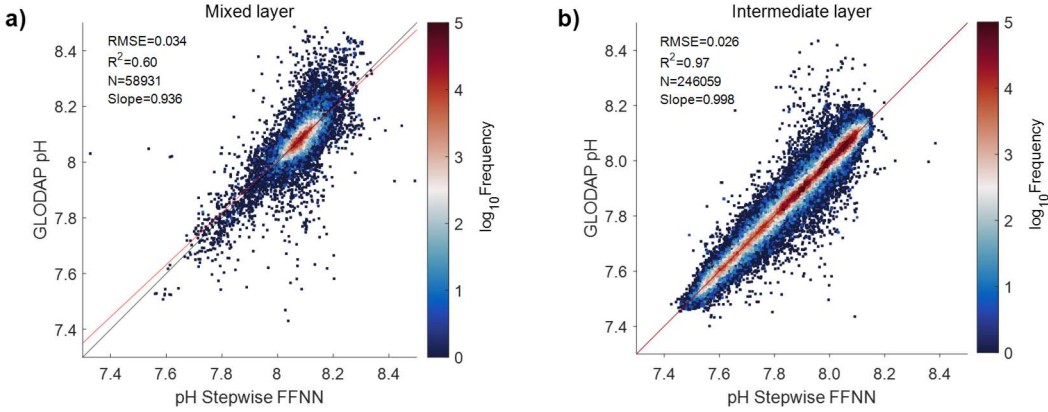

**Figure 4: Comparison between FFNN pH and GLODAP pH measurements.** a): mixed layer from the surface to mixed layer depth; b): intermediate layer from mixed layer depth to 2000 m. Black lines: the y=x line; red lines: the linear regression between GLODAP pH and Stepwise FFNN pH (Lauvset et al., 2023); Slope: slope of the linear regression.

The RMSE between the FFNN pH and GLODAP pH measurements at most grids were lower than 0.03 (Figure 5a). The performance of FFNN was relatively better in the temperate oceans, with the RMSE lower than 0.02 at some temperate grids. However, relatively higher RMSE was found in the equatorial and polar oceans, especially in the eastern equatorial Pacific, the near-polar North Pacific, and the northwest Indian Ocean. The RMSE was relatively lower in regions with concentrated GLODAP measurements, such as the near-polar North Atlantic, south Atlantic, and south Indian Ocean.

Due to the higher seasonal and interannual variability of seawater pH near the surface ocean, the RMSE decreases with depth in all basins (Figure 5b). At the surface ocean, the RMSE between the FFNN pH and the GLODAP pH measurements was 0.044. The RMSE fluctuates between 0.032 and 0.048 at the subsurface 0-200 m. The RMSE between the FFNN pH and the GLODAP pH measurements decreased rapidly from the 200 m depth. In the global ocean 1500-2000 m depth, the global RMSE was lower than 0.015. While at 2000 m depth, the global ocean RMSE at 2000m was 0.013, with the higher RMSE in the Arctic Ocean and the lower in the Southern Ocean. The vertical distribution of the RMSE and statistical distribution of pH difference in different basins all suggested a relatively higher reconstruction error in the mixed layer than in the intermediate layer (Figure 5d). The vertical difference of RMSE between the mixed layer and intermediate layer was most notable in the Arctic and Indian Ocean, where the RMSE at different depths was also higher than the other basins. The RMSE in the surface Arctic Ocean was higher than 0.10 and decreased rapidly to 0.025 by 450 m depth. On the contrary, the RMSE of the surface Indian Ocean was 0.018, but increased to 0.053 by 80 m depth and then decreased continuously with depth. The high RMSE of subsurface oceans is because there are almost no GLODAP pH measurements in the entire Indian Ocean at 50-150 m depth. The RMSE in different years also suggested the notable influence of pH measurement amount on the FFNN reconstruction errors. The RMSE in the early years was relatively higher than in recent years, while the number of GLODAP measurements increased with the years (Figure 5c).

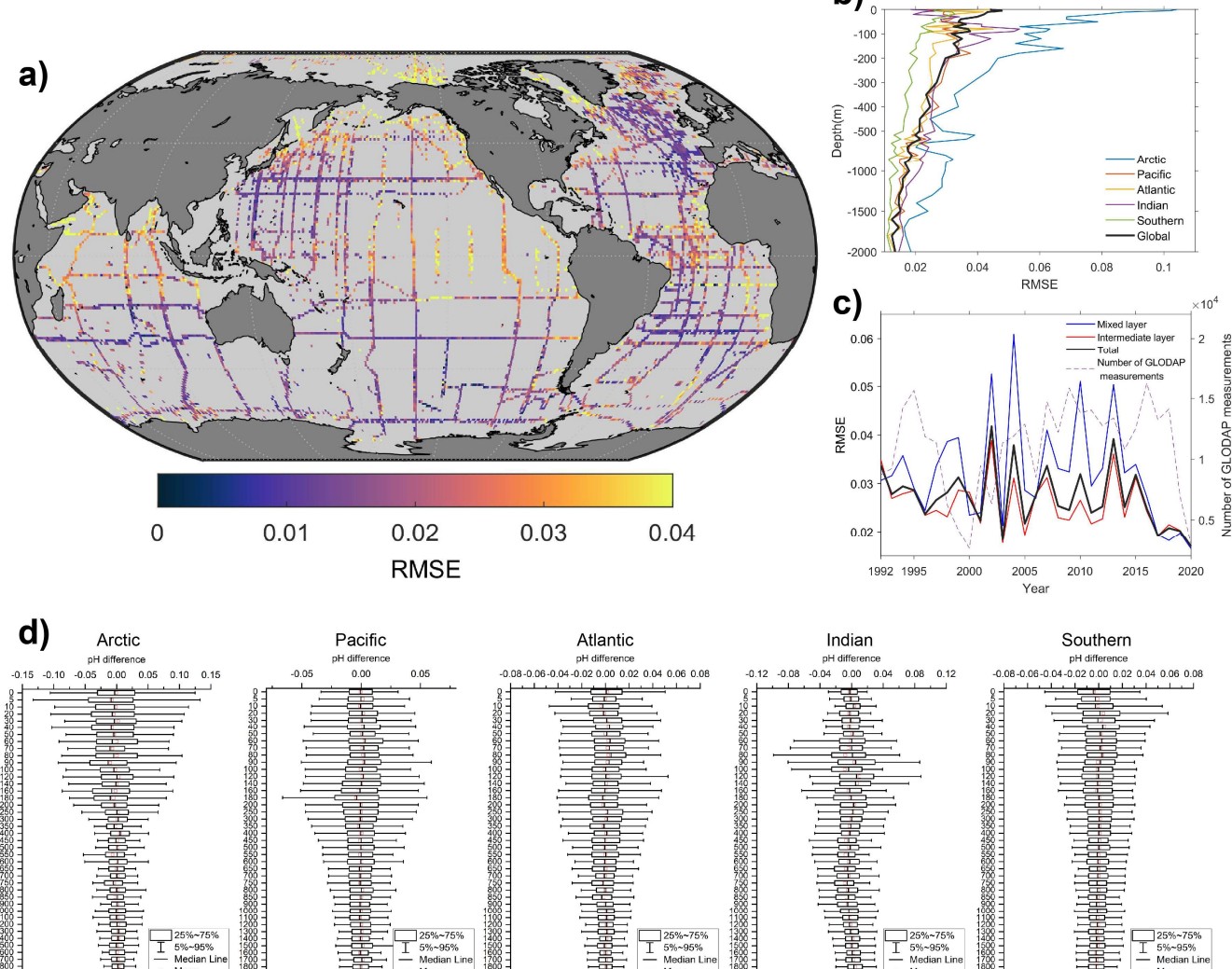

**Figure 5: Distribution of RMSE between FFNN pH values and GLODAP pH measurements.** a): global spatial distribution of RMSE between FFNN pH and GLODAP pH measurements at 0-2000 m (Lauvset et al., 2023); b): basin average RMSE at different depth; c): temporal distribution of global RMSE; d): Statistical distribution of pH difference between reconstructed pH values and GLODAP pH measurements in each basin.

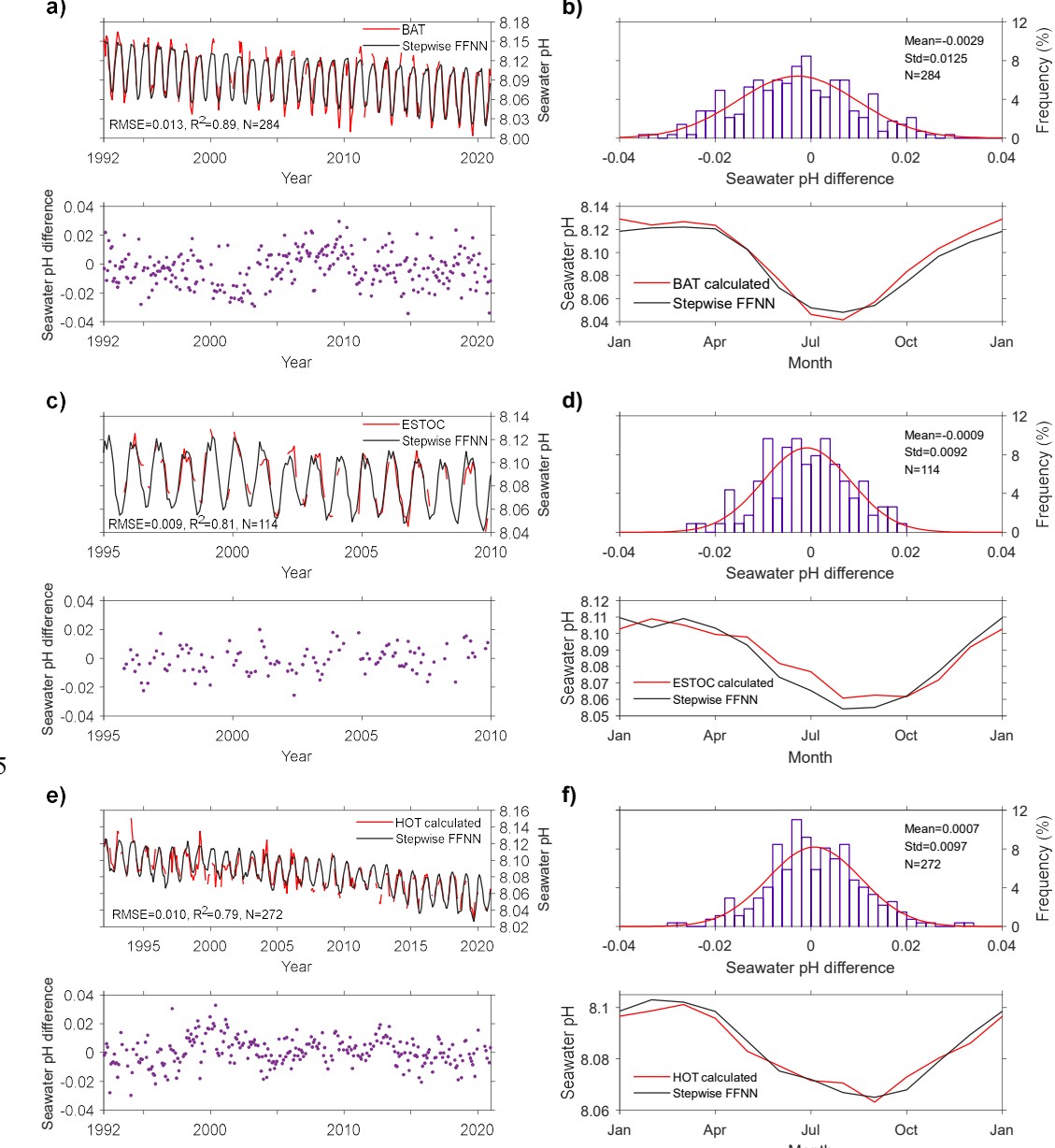

**Figure 6: Comparison between FFNN pH and time-series measurements.** a-b): pH value, pH difference and its distribution, pH seasonal variability of FFNN result and time series measurements at the BAT station; c-d): the ESTOC station; e-f): the HOT station.

The Stepwise FFNN pH product showed variability of seawater pH close to the independent time series observations in the surface ocean from HOT, ESTOC, and BAT stations (Figure 6). At the BAT station, the RMSE between the reconstructed pH and time series observations was 0.013. The surface seawater pH of our Stepwise FFNN product decreased by $0.0017 \pm 0.0007$ yr$^{-1}$ on average during the past three decades at the BAT station, close to the $-0.0018 \pm 0.0001$ yr$^{-1}$ of BAT time series observations in the same period (Bates et al., 2020). At the ESTOC station, the Stepwise FFNNN product and time series observations were also well consistent, with the RMSE of 0.009 and a similar long-term trend (González-Dávila et al., 2010). The RMSE between the Stepwise FFNN product and HOT time series observations was also 0.010, and the long-term trends of the Stepwise FFNN pH product was $0.0018 \pm 0.0004$ yr$^{-1}$, consistent with the HOT time series observations. Although at the BAT station, the Stepwise FFNN product suggested a smaller seasonal change scale than the time series observations, the seasonal patterns of surface seawater pH were consistent between the Stepwise FFNN product and time series observations at all three stations. The extreme values not reconstructed by the FFNN are mainly observed at the BAT station near 2010 and at the HOT station near 2000 during La Niña events, and at the HOT station before 2000 during El Niño events. Differently, the extreme values not reconstructed by the FFNN are less observed at the ESTOC station, where the surface pH did not notably

fluctuate during El Niño/La Niña events. It can be inferred that the extreme values not reconstructed by the FFNN may be due to its underestimating of the impact of El Niño/La Niña events on pH of certain temperate areas. Compared to previous surface ocean seawater pH product, which were derived from reconstructed DIC, TA, or $p$CO$_2$ products, the Stepwise FFNN product

was consistent with the pH trend from the majority of time series stations (Table 4). The long-term pH trend of our product at the ESTOC station was slower than other gridded products, but the result is still close to the -0.0016 ± 0.0001 yr$^{-1}$ of real observations. In the Irminger Sea station, the FFNN pH trend was notably faster than the result of time series observations. However, differences in pH trend among pH products were most remarkable in this station. On the global scale, the pH trend of our FFNN product is -0.0015 ± 0.0002 over the period from 1992 to 2020. There is no significant difference between our

FFNN product, the CMEMS product, and the Copernicus product under the current uncertainty.

**Table 4: Comparison of surface acidification rate with previous product in different time series stations and on a global scale.**

| Stations | Period | Time series observation | Stepwise FFNN (This study) | JMA (Iida et al., 2021) | CMEMS (Chau et al., 2024) | OS-ETHZ (Gregor et al., 2021) | Copernicus (Copernicus Marine Service, 2020) |
|---|---|---|---|---|---|---|---|
| BAT | 1992~2020 | -0.0018 ± 0.0001 | -0.0017 ± 0.0007 | -0.0018 ± 0.0002 | -0.0018 ± 0.0002 | -0.0018 ± 0.0002 | - |
| ESTOC | 1995~2010 | -0.0016 ± 0.0001 | -0.0014 ± 0.0005 | -0.0022 ± 0.0003 | -0.0020 ± 0.0002 | -0.0017 ± 0.0003 | - |
| HOT | 1992~2020 | -0.0018 ± 0.0001 | -0.0018 ± 0.0004 | -0.0020 ± 0.0001 | -0.0021 ± 0.0001 | -0.0019 ± 0.0001 | - |
| Iceland Sea | 1992~2019 | -0.0020 ± 0.0004 | -0.0028 ± 0.0002 | -0.0030± 0.0003 | -0.0015 ± 0.0002 | -0.0020 ± 0.0002 | - |
| Irminger Sea | 1992~2019 | -0.0025 ± 0.0004 | -0.0022 ± 0.0002 | -0.0027 ± 0.0002 | -0.0017 ± 0.0003 | -0.0016 ± 0.0003 | |
| DYFAMED | 1998~2017 | -0.0010 ± 0.0008 | -0.0005 ± 0.0003 | - | -0.0017 ± 0.0003 | -0.0023 ± 0.0004 | |
| Global | 1992~2020 | - | -0.0015 ± 0.0002 | -0.0018 ± 0.0000 | -0.0017 ± 0.0004 | -0.0018 ± 0.0000 | -0.0017 ± 0.0002 |

(the trend from different products for comparison were recalculated based on data during same period noted in the second column; Stepwise FFNN product: reconstructed from pH measurements with 1°×1° and monthly resolution from 1992 to 2020, covering global open ocean 0-2000 m; JMA product: reconstructed from DIC and Alk with 1° and monthly resolutions from 1990 to 2022, covering global surface ocean

except a portion of the Arctic; CMEMS product: reconstructed from $p$CO$_2$ and Alk with 1° or 0.25° resolutions and monthly resolution from 1985 to 2021, covering global surface ocean except a portion of the Arctic; OS-ETHZ product: reconstructed from $p$CO$_2$ and Alk with 1° and monthly resolutions from 1982 to 2022, covering global surface ocean except the Arctic; Copernicus product: mean sea water pH time series and trend from Multi-Observations Reprocessing, from 1985 to 2021)

Compared with the time series data below the surface, the FFNN pH was close to the pH observations at upper few hundred meters in the BAT and HOT station (Figure 7). However, higher RMSE and larger ranges of pH difference were observed at 500-1500 m in the BAT station and below 300 m in the HOT station. This may be due to the sparser GLODAP observations used to train the FFNN model in these areas. Additionally, as depth is used as a pH predictor, in the validation based on the GLODAP dataset the FFNN pH values used for validation were outputted at the same depth of the GLODAP

observations. When comparing FFNN pH with independent time series observations, differences in depth between the pH product and the observations can amplify the calculated pH difference and RMSE. For example, the FFNN pH product was reconstructed at depths of 1800 m and 2000 m in the bottom. If the time series observation is at 1910 m depth, it will be compared with the FFNN pH value at 2000 m in the independent validation. This depth difference significantly increases the pH error in validation based on independent data. Despite higher RMSE at certain depths, the RMSE at most depths in the

deep areas of the BAT station and DYFAMED station is below 0.03, indicating that the notable deviations may only occur at local scale.

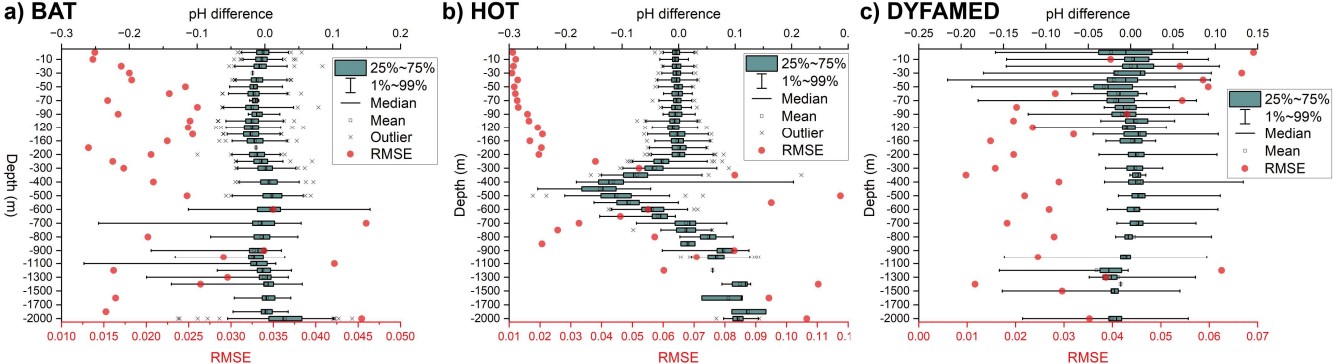

**Figure 7. RMSE and pH difference between FFNN pH and time series observations at different depths.** a) BAT station at 31°50' N, 64°10' W based on data from 1992 to 2020; b) HOT station at 22° 45' N, 158° 00' W based on data from 1992 to 2020; c) DYFMED station at 42.3°N, 7.5° E based on data from 1998 to 2017.

### 3.1.2 Validation based on BGC-Argo float pH measurements

Comparison with time series observations in deeper oceans suggested that the distribution of pH reconstruction errors with depth varies notably across different stations. To better assess the performance of FFNN in the reconstruction of pH at different depths, the FFNN reconstructed pH was further evaluated by comparing with independent BGC-Argo delayed-mode pH-adjusted data with quality control flag 1 at various depths (Argo, 2024), with spatial positions showing in Figure S6. Different from the validation results based on the GLODAP dataset, the RMSE between FFNN pH and BGC-Argo pH data in the intermediate layer is 0.051, higher than 0.035 in the mixed layer (Figures 8a and 8b). In both the mixed layer and intermediate layer, most samples were evenly distributed around the y=x line. However, in the intermediate layer, some samples were slightly offset and distributed below the y=x line, which may be the main reason for the notably higher RMSE between FFNN pH and BGC-Argo pH data in the intermediate layer. Overall, there is a good linear correlation between FFNN reconstructed pH and independent BGC-Argo pH data, with R² values of 0.73 and 0.84 in the mixed layer and intermediate layer, respectively.

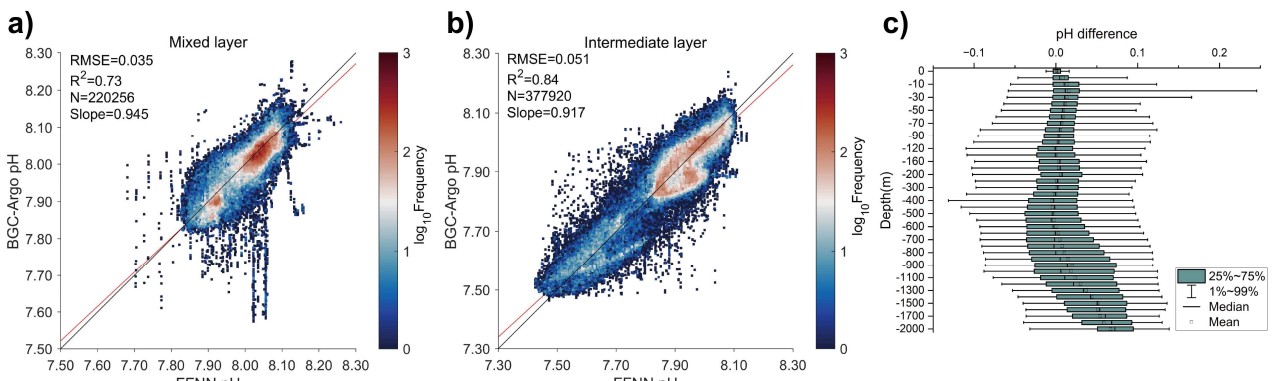

**Figure 8. Difference between FFNN pH and BGC-Argo floats pH.** a) comparison between FFNN pH and BGC-Argo floats pH in the mixed layer; b) comparison between FFNN pH and BGC-Argo floats pH in the intermediate layer; c) Statistical distribution of pH difference (FFNN pH minus BGC-Argo floats pH) at different depth levels. FFNN pH: pH data reconstructed in this work; BGC-Argo pH: pH data from BGC-Argo data from France Coriolis GDAC (Argo, 2024).

The distribution of pH differences between FFNN pH and BGC-Argo pH data at different depths reveals a relatively smaller biases above 500 m (Figure 8c). However, below 500 m, the bias between FFNN pH and BGC-Argo pH data increases with depth and was the most remarkable at 2000 m. Comparing the pH bias calculated based on BGC-Argo dataset and GLODAP dataset, it is evident that only the bias between FFNN pH and BGC-Argo pH data tends to be more notable in deep areas except the Pacific Ocean (Table 5). In contrast, greater biases between FFNN pH and GLODAP pH occur mainly in the

surface layer, with the most in the surface Indian Ocean. This disparity in distribution patterns between biases based on BGC-Argo dataset and GLODAP dataset is most remarkable in the Southern Ocean, where the bias between FFNN pH and GLODAP pH is nearly zero below 1000 m, compared to biases between FFNN pH and BGC-Argo pH data ranging from 0.040 to 0.068. These differences between FFNN pH and BGC-Argo pH data are primarily attributed to the discrepancies between GLODAP dataset and the BGC-Argo dataset in the deep ocean, as our product was based on the GLODAP dataset and small biases with GLODAP pH were observed in the deep ocean.

**Table 5. pH bias by area and depth computed with BGC-Argo and GLODAP dataset.**

|  | Area |  | 0-50 m | 50-200 m | 200-500 m | 500-1000 m | 1000-1500 m | 1500-2000 m |
|---|---|---|---|---|---|---|---|---|
| Pacific | BGC-Argo | bias | 0.028 | 0.016 | -0.003 | -0.013 | 0.027 | -0.004 |
|  |  | N* | 16433 | 34708 | 36431 | 19840 | 8772 | 3565 |
|  | GLODAP | bias | -0.001 | -0.001 | 0.000 | 0.000 | 0.000 | -0.001 |
|  |  | N | 18687 | 26629 | 22746 | 24843 | 12613 | 13817 |
| Atlantic | BGC-Argo | bias | 0.018 | 0.019 | 0.013 | -0.021 | 0.031 | 0.068 |
|  |  | N | 3285 | 6832 | 7152 | 3565 | 1622 | 1288 |
|  | GLODAP | bias | 0.000 | 0.000 | -0.001 | -0.001 | 0.000 | 0.000 |
|  |  | N | 11808 | 15894 | 14330 | 18056 | 10686 | 11780 |
| Indian | BGC-Argo | bias | 0.023 | 0.034 | 0.025 | -0.022 | 0.000 | 0.036 |
|  |  | N | 407 | 916 | 920 | 491 | 241 | 57 |
|  | GLODAP | bias | -0.006 | -0.001 | -0.003 | -0.004 | -0.004 | -0.001 |
|  |  | N | 3145 | 5397 | 5124 | 5276 | 3457 | 3421 |
| Southern | BGC-Argo | bias | 0.008 | 0.000 | 0.001 | 0.015 | 0.040 | 0.068 |
|  |  | N | 66436 | 130563 | 135817 | 72564 | 27579 | 18692 |
|  | GLODAP | bias | 0.004 | 0.001 | 0.001 | 0.000 | 0.000 | 0.000 |
|  |  | N | 7983 | 12268 | 10457 | 10341 | 6169 | 5800 |
| Global | BGC-Argo | bias | 0.012 | 0.004 | 0.001 | 0.008 | 0.036 | 0.057 |
|  |  | N | 86561 | 173019 | 180320 | 96460 | 38214 | 23602 |
|  | GLODAP | bias | -0.001 | 0.000 | 0.000 | -0.001 | -0.001 | 0.000 |
|  |  | N | 46415 | 66635 | 57491 | 62447 | 34994 | 37008 |

(*: N is the number of BGC-Argo or GLODAP samples used to compute the biases.)

## 3.2 Gridded pH product

### 3.2.1 Spatial pH distribution

The spatial distribution of long-term average seawater pH in the Stepwise FFNN product suggests the lowest surface seawater pH in the equatorial Pacific with an average value near 8.00 (Figure 9a), which is in good agreement with the surface seawater pH range of 7.91-8.12 observed in the equatorial Pacific in recent decades (Sutton et al., 2014). The upwelling transporting the deep water with high dissolved inorganic carbon and low pH to the surface was the main driver. The equatorial Indian Ocean and the equatorial Atlantic also show a low surface pH of about 8.05, consistent with the distribution patterns of the GLODAP pH climatology (Lauvset et al., 2016). The highest surface pH is found in the Atlantic sector of the Arctic Ocean, where the average surface pH was around 8.15 during the past three decades. Besides, the average surface pH in temperate oceans is relatively higher, such as the south Indian and south Atlantic Oceans. In the temperate Pacific Ocean, differences in surface pH levels were observed between the west and east in both our product and GLODAP pH climatology, which may be caused by the spread of eastern equatorial seawater with extremely low pH. At the deeper depth of 1000 m, the spatial

distribution pattern of FFNN pH product is generally consistent with the GLODAP climatology, despite still some disturbance of bad FFNN performance along the SOM province boundary and the higher FFNN pH in the Southern Ocean.

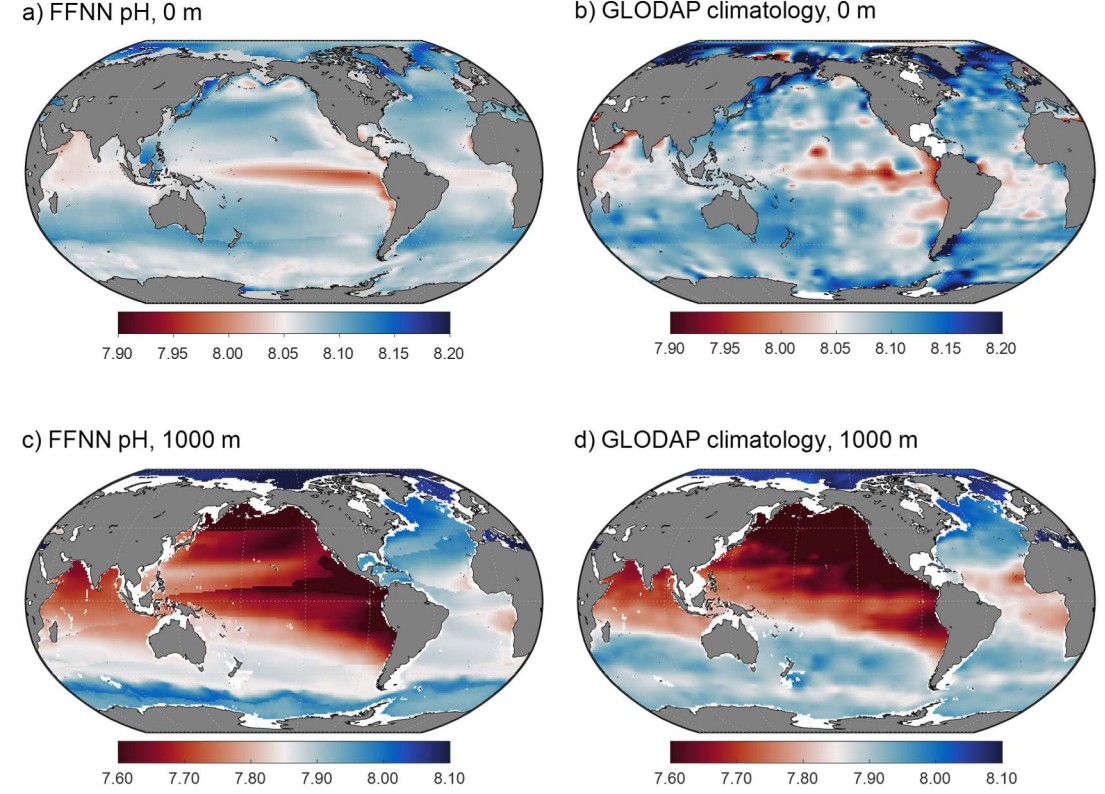

**Figure 9: Average pH distribution from FFNN pH product and GLODAP climatology normalized to the year 2002.** The GLODAP climatology data is from Lauvset et al., 2016.

The vertical distribution of average pH in the proposed product showed a notable pH decrease with increasing depth in the upper 500 m of different basins (Figure 10). The seawater pH was the lowest at nearly 500 m and rose with increasing depth at 500-2000 m in the Pacific and Atlantic oceans. The distribution pattern of seawater pH in the Indian Ocean was similar to that in the South Pacific, with the lowest seawater pH appearing near 1000 m. The subsurface seawater with low pH in the Atlantic Ocean and Indian Ocean was mainly concentrated in the equatorial region. In contrast, subsurface seawater with low pH in the Pacific Ocean appeared in subpolar and equatorial regions. The overall distribution pattern of the reconstructed pH is in good agreement with previous research (Lauvset et al., 2016; Lauvset et al., 2020). It can be concluded that the FFNN fitted the relationship between GLODAP seawater pH and its predictors well, and the proposed pH product has good accuracy.

Based on the pH predictors selected by the Stepwise FFNN algorithm, differences in processes driving pH variability were identified between the mixed layer and intermediate layer in most provinces. In the mixed layer, surface ocean $p$CO$_2$ was identified as the most informative predictor in many provinces, followed by temperature and nutrient concentration. This suggests that the CO$_2$ exchange between surface ocean and atmosphere is the primary driver of pH variability, followed by biological CO$_2$ utilization and seasonal changes in seawater temperature. In contrast, phosphate was identified as the most informative predictor in the intermediate layer, followed by temperature and depth. This suggests that the primary process driving pH variability is the remineralization of organic matter, converting organic carbon into inorganic forms and also releasing nitrogen and phosphorus. Given the notably smaller seasonal temperature changes in the intermediate layer compared to the mixed layer, the selection of temperature as an important pH predictor may indicate a notable influence of ocean warming on seawater pH variability. Additionally, depth was also selected as an important predictor in the intermediate layer. The observed pattern of seawater pH decreasing with increasing depth in most provinces, as suggested by the constructed pH product, may be the main reason.

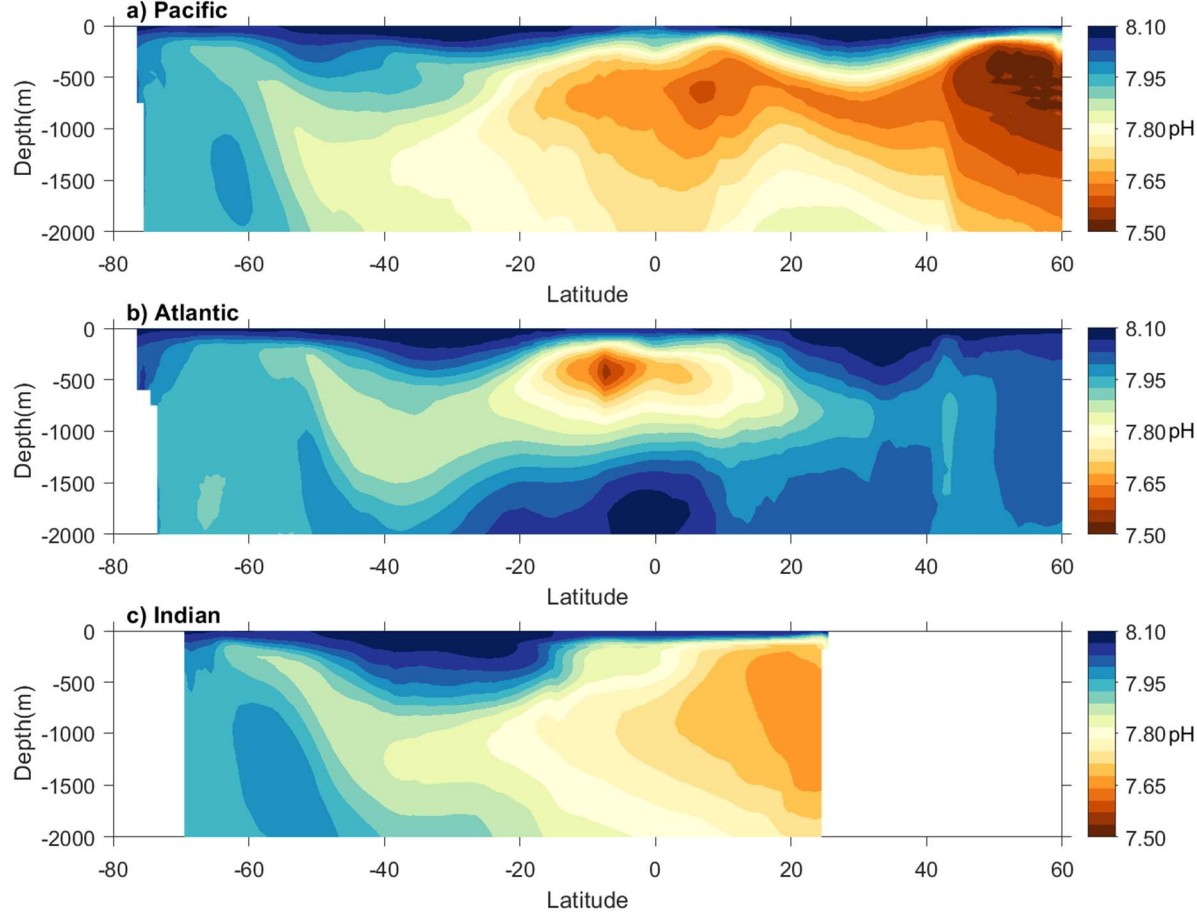

**Figure 10: Climatological vertical distribution of zonal average FFNN pH in main basins.** The pH values shown at each latitude were averaged from pH values across all longitudes within each major basin.

### 3.2.2 Uncertainty

As described in the method section, the FFNN pH was converted to [H$^+$] to calculate the regional RMSE of [H$^+$] between FFNN results and GLODAP measurements, and then the RMSE of [H$^+$] in each SOM province was used to calculate the pH product uncertainty caused by the construction algorithm (Equation 2). Due to higher reconstruction errors, the pH product uncertainty is relatively higher near the surface (Figure 11). The uncertainty is generally lower than 0.02 at depths from 500 m to 2000 m, except for some regions near the SOM province and vertical boundary. Although we have used a cross-boundary method to improve the FFNN performance near the SOM and vertical boundary, there are still some discontinuity problems and relatively higher uncertainty. This is because the pH values on two sides of the SOM boundary were reconstructed from two different FFNN models, which were trained with different samples and used different predictors. If one of the FFNN models experienced a worse performance due to insufficient training samples or predictors, the pH values on two sides of the SOM boundary will still differ notably, that is, discontinuity along the boundary. Therefore, the analyze on a regional scale based on pH values near SOM boundaries should be more cautious when using our product. In addition, the equatorial and polar regions show an uncertainty higher than 0.04. This is because the FFNN performance tends to be worse in regions with the highest and lowest pH levels than in regions where pH values are near the average level. Especially in the Arctic Ocean, the pH measurements are much sparser leading to the highest reconstruction error and pH uncertainty. Therefore, the proposed pH product should be cautiously used in regional analysis near the boundaries or equatorial and polar regions.

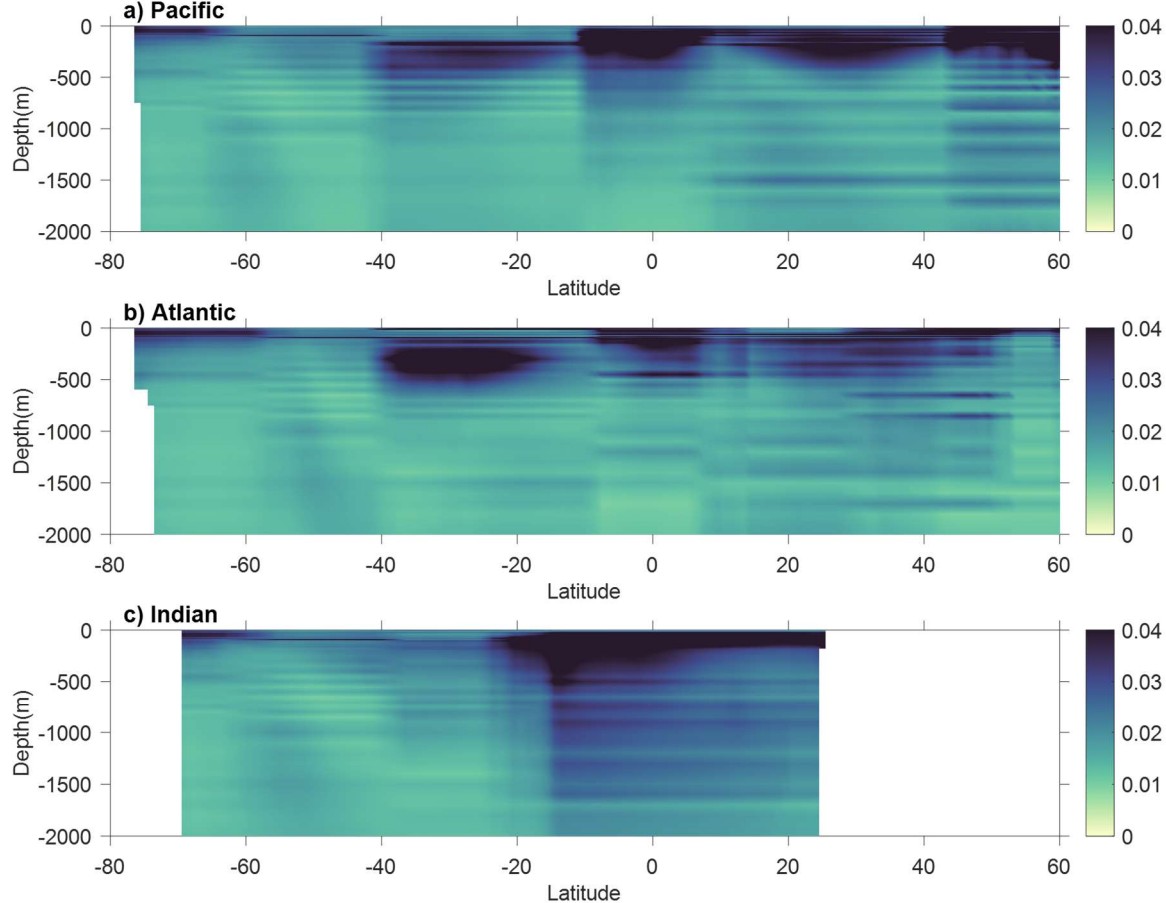

**Figure 11: Uncertainty of FFNN pH product in main basins.**

## 4 Data availability

The materials used in this research including the gridded seawater pH product (NetCDF files for all individual years), MATLAB code for reconstruction and validation, and other materials (available as .m or .mat files) are available from the Marine Science Data Center of the Chinese Academy of Sciences at http://dx.doi.org/10.12157/IOCAS.20230720.001 (Zhong et al., 2023). The used pH measurements are available from GLODAP (https://glodap.info/index.php/merged-and-adjusted-data-product-v2-2023/, Lauvset et al., 2023), data products used for predictors are available from references listed in Table 1.

## 5 Conclusions

Quantifying the global seawater pH variability is important for understanding the future responses of oceans on the uptake of anthropogenic $CO_2$. A four-dimensional global seawater pH product covering depths from the surface to 2000 m and years from 1992 to 2020 was reconstructed in this work. This product serves as a reference for guiding acidification surveys by providing a general understanding of acidification process at different depths on a basin scale and indicating areas with potential fast or slow acidification rates. Additionally, the pH product brings insights into acidification research and can be used to analyze the influence of specific ocean processes on acidification rates and the broader impacts of acidification on a large scale when direct observations are unavailable. However, caution should be exercised when using this product for regional analyses at a small spatial scale. The analysis of pH RMSE and uncertainty suggested that the proposed pH product remains limited in equatorial and polar regions and along the SOM boundary lines. This limitation was caused by sparse

measurements and method disadvantages, which can be mitigated in future improvement works. Potential improvement may be achieved by increasing more predictor products to capture the pH drivers, testing more machine learning algorithms, and accumulating more seawater pH observations. Besides, the method used to reconstruct the pH product can be applied in reconstruction of global fields of other ocean chemical variables, such as nutrients, particulate organic carbon, and dissolved inorganic carbon. The global field of these variables may further improve the pH product accuracy, as climatological products of these variables were used as pH predictors and lacked interannual variability information. Overall, decreasing seawater pH will influence the metabolism of marine organisms and result in notable changes to the marine ecosystem. The discrete observations may be insufficient to support research on large scales. With the machine learning method in this work, the discrete pH measurements were mapped to global gridded fields to fill the unsampled areas. Our product can be used for analysis of seasonal to decadal and regional to global pH variability, to break through the limitation of discrete observations.

**Acknowledgments**

We thank the data support of the Marine Science Data Center and Public Technical Service Center, Institute of Oceanology, Chinese Academy of Sciences. We thank GLODAP for sharing the pH observation data. We thank BGC-Argo for sharing the pH float data. These data were collected and made freely available by the International Argo Program and the national programs that contribute to it (http://www.argo.ucsd.edu, http://argo.jcommops.org). The Argo Program is part of the Global Ocean Observing System. We thank the support by the National Natural Science Foundation of China (grant nos.42176200); National Key Research and Development Program (2022YFC3104305); Laboratory for Marine Ecology and Environmental Science, Qingdao National Laboratory for Marine Science and Technology (LSKJ202204001, LSKJ202205001); Shandong Province and Yantai City Talent Programs; and Science Fund for Creative Research Groups of the National Natural Science Foundation of China (42221005).

**Author contributions:**

Data product collection: Baoxiao Qu, Yanjun Wang, Bin Zhang; Data product synthesize: Jun Ma, Qidong Wang, Jianwei Xing; Methodology: Guorong Zhong, Jiajia Dai, Liqin Duan, Ning Li; Model improve: Xuegang Li, Jinming Song, Huamao Yuan; Writing – original draft: Guorong Zhong, Xuegang Li; Writing – review & editing: Jinming Song, Fan Wang, Lijing Cheng.

**Competing interests:** Authors declare that they have no competing interests.

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
