# Peer review of "A global monthly 3D-field of seawater pH over 3 decades: a machine learning approach"

_Earth System Science Data, 2024_

## Referee Comment (RC1)

**Review of: "A global monthly field of seawater pH over 3 decades: a machine learning approach" by G. Zhong et al.; submitted to Earth System Science Data**

**Context and general comment :**

The continuous absorption of anthropogenic CO2 by the ocean leads to ocean acidification, which threatens marine ecosystems. While the acidification rate has been extensively documented at the surface, data for deeper waters remain limited. Zhong et al. address this gap by presenting a comprehensive, monthly, four-dimensional, 1°×1° gridded global seawater pH dataset, covering the years 1992 to 2020 and depths from the surface to 2000 meters.

This dataset was developed using machine learning algorithms trained on pH observations from the Global Ocean Data Analysis Project (GLODAP). The methodology employed is a three-step process: 1) self-organizing map neural network for bioregionalization, 2) a stepwise algorithm for predictors selection, and 3) feed-forward neural networks (FFNN) for non-linear regression. The resulting pH product is a valuable resource for studying subsurface ocean acidification and for validating or initializing biogeochemical models. The product is made publicly available through the Marine Science Data Center of the Chinese Academy of Sciences.

Overall, the article is well-written and the figures are clearly presented.

Despite the significance of this new 3D pH product for the scientific community, the article has some notable shortcomings. There is a lack of details in the methodology section, which makes it challenging to fully evaluate the robustness of the method and comprehend the implications involved.

**Specific Comments:**
**Title:**
It may be valuable to the reader to add the information that the estimations are depth-resolved, resulting in a 3D product, which is the principal novelty of this methodology.

**Abstract:**
- *Lines 15-17 :* "Here, we present a monthly four-dimensional 1°×1° gridded product of global seawater pH, derived from a machine learning algorithm trained on pH observations at total scale and in-situ temperature from the Global Ocean Data Analysis Project (GLODAP)." : The role of temperature in the methodology is unclear. Even after reading the entire paper, the specific role of temperature compared to other inputs remains ambiguous.
- *Line 18 :* I suggest rephrasing the method description for clarity. Consider stating: "A three-step machine learning-based algorithm was used..."
- *Line 19 :* The term "stepwise" may not be clear to the readers. Consider elaborating or using a more descriptive term.

**Introduction:**
- The introduction appears to be missing some crucial references. For example, it would be beneficial to acknowledge that the methodology is inspired by the work of Landschützer et al. (2014) and references following ; i.e. a SOM-FNN approach. Additionally, it is important to mention that this SOM-FNN approach has already been applied to the 3D reconstruction of DIC by Keppler et al., 2020

(https://agupubs.onlinelibrary.wiley.com/doi/10.1029/2020GB006571). These references should be cited to provide a more comprehensive background.

- L*ine 52 :* where existing pH surface products are listed, there is a lack of references and details. It is crucial to include comprehensive citations of existing products (e.g. LSCE-FFNN), especially those used for comparison in Table 4 (+ the missing ones, see my comment below).

- The paper does not explain why the product spans the period 1992-2020. It would be helpful to provide a rationale for this timeframe and discuss why it does not cover a longer period, both in the past and up to the present (i.e., year-1).

- L*ine 55:* The reference to GLODAP by Lauvset et al. (2022) refers to GLODAPv2.2022 and should be cited as such throughout the article (instead of 'GLODAP'). Additionally, if the authors re-run their model during the review process, it is suggested to use the latest version of GLODAP, i.e., GLODAPv2.2023. The updated reference can be found here: https://essd.copernicus.org/articles/16/2047/2024/essd-16-2047-2024-discussion.html

**Methods:**

*Line 60 :* It is unclear how temperature is used in the methodology. Additional explanation is needed to clarify its specific role and contribution.

*Lines 69-73 :*The inclusion of other indices, such as the Northern Oscillation Index, should be considered.

*Lines 71-73 :* The LSCE-FFNN product provides total alkalinity and DIC data monthly from 1985 to year-1: data available from the Copernicus Marine Service (https://data.marine.copernicus.eu/product/MULTIOBS_GLO_BIO_CARBON_SURFACE_REP_015_008/description). If authors refer to 3D products, then it has to be clearly mentioned. For 3D estimations, DIC is available monthly from 2004 to 2019 from MOBO-DIC (Keppler et al., 2020).

*Section 2.2 :*

- *Lines 103-105 :* The rationale for using these specific parameters to define bioregions needs clarification, especially if they are not significant in the stepwise algorithm for determining important parameters in relation to pH.

- *Line 106 :* The criteria and process for merging provinces with fewer than ten connected grids or less than 100 GLODAP pH measurements should be rephrased and/or detailed because it is unclear.

- *Line 107 :* The need for manual subdivision of provinces separated by continents requires further explanation. Why not using same bioregion even if it is not in the same ocean, it is possible that the uderlying processes are equivalent and so that the FFNN will be performant in both basin ?

- *Lines 112-114 :* The sentence on the division of ocean areas into different layers also requires further details on which drivers are important for each layer as it is stated that drivers differ depending on the layers. Moreover, following this statement, why using the same bioregions for deeper layers ?

*Section 2.3 and Table 1 :*

- The choice of a single-layer FFNN instead of a multi-layer network should be justified. Has this been tested ?

- The use of sin(Lat) as a predictor is questionable since latitude is not circular.

- Clarify how depth is used as a predictor and whether it corresponds to the depth of retrieval of the output or if the FFNN estimates X values for X depth levels.

- The choice of the ECCO2cube92 model should be discussed and better supported by citations in the text.
- MEI should be defined as the Multivariate ENSO Index.
- Adding a column to Table 1 to indicate which process each variable is associated with would be informative.
- Consider using merged satellite ocean color data products like OC-CCI or GlobColour for longer time series would help for future usability and sustainability.
- Provide details on how the most informative parameters were chosen and how hyperparameters (architecture, number of neurons) were handled in this stepwise process.
- Clarify how co-correlation among selected predictors was removed in the stepwise FFNN selection procedure.
- The sentence regarding additional FFNNs trained with predictors in Table S1 for polar areas and periods before August 2002 needs clarification.
- Discuss Tables 2 and 3 scientifically in the Results section to highlight important processes driving pH variability.
- Figure 3 is extremely difficult to understand and should be clarified or redesigned.
More generally, the section 2.3 is currently unclear and needs to be rewritten with more detailed explanations.

*Section 2.4:*
*Line 191:* The paragraph is unclear. The statement, "Therefore, the uncertainty of our pH product was directly estimated from the FFNN pH predicting errors, instead of synthesizing the inherent uncertainty of each used predictor product," needs further clarification. How was this done?

*Section 3.1:*
- *Line 214 :* This interpretation might be overstated. The broader value range likely contributes to a better model fit, and pH values exhibit less variability at depth.
- *Figure 4 :* Authors might add the slopes of the linear regression to the statistics.
- *Line 235 :* The impact of the Oxygen Minimum Zone (OMZ) on the product should be discussed more in details.
- *Figure 5 :* This figure requires re-arrangement. The map should be larger, and pH differences against depth should be plotted with depth as the y-axis, as is more common for reading profiles. Additionally, including seasonal variability for each major basin along with yearly variability would be beneficial.
- In the validation section, it would be valuable to compare the global scale trend with the Copernicus Marine Service data: https://marine.copernicus.eu/access-data/ocean-monitoring-indicators/global-ocean-acidification-mean-sea-water-ph-time-series.
Moreover, it would be interesting to add comparison against qualified pH data from BGC-Argo dataset.
- *Figure 6 + text:* Comparing to other available pH time series would be interesting. These are listed in the recent ESSD paper by Lange et al. (2024): https://essd.copernicus.org/articles/16/1901/2024/. For instance, the Mediterranean Sea, where data from GLODAP V2 are very scarce, could be validated against the Dyfamed pH time series.
- *Figure 6 :* Discuss the extreme values not reconstructed by the FFNN in the text.

- *Line 254 :* Chau et al. (2022) may not be the best reference, as they are also model (ML)-based.

- *Line 261* : Describe in what specific ways the product differs from other products.

- *Table 4:* More products could be compared, such as Jiang et al. (2022): Remote Sensing of Global Sea Surface pH Based on Massive Underway Data and Machine Learning (https://doi.org/10.3390/rs14102366). Additionally, some products compared here have not been previously cited in the article (refer to the comment on the introduction). The effect of the different time ranges of the different products on the computation of trends should also be analyzed and discussed.

*Section 3.1.2 and Figure 7:* Not sure whether this paragraph and figure are necessary.

*Section 3.2:*

- *Lines 301-304 :* This issue is problematic and should be discussed in more details for the user. Additionally, the significant differences between the GLODAP climatology and this product at 1000 m in the Southern Ocean should be discussed/addressed.

- *Figure 9 :* The longitude of the zonal average should be specified in the caption and/or the text.

- *Section 3.2.2 :* The discontinuity problem requires more discussion, both methodologically (explaining why this issue occurs despite the use of the cross-boundary method) and in terms of implications for users. If local uncertainties are available, they should be included in the NetCDFs.

*Section 5:*

Authors should provide more concrete examples of applications for their product in the Conclusion.

**Typos :**

- *Line 78 :* was converted to a 1°×1° resolution by averaging  0.25° grids into one 1° grid

- *Line 84 :* *

- *Line 116 :* Therefor**e**

**Data:**

I encountered an error when attempting to open the NetCDF file using R. The error message was as follows:

```
Dans nc_open("/home/user/Data/2012.nc") :
  WARNING file /home/user/Data/2012.nc is not compliant netCDF; variable pH  is
numeric but has a character-type missing value! This is an error!  Compensating,
but you should fix the file!
```

Although I didn't receive any warnings when using xarray with Python, this issue should be addressed to ensure compatibility with other tools. Additionally, when opening the dataset with xarray and attempting to plot it using the library's functions, I noticed that the longitude and latitude are reversed (not in the name), and the longitude is plotted on the y-axis. To enhance user-friendliness when using Python tools, it would be beneficial to adjust the format accordingly.

Regarding the availability of MATLAB code, I am not a MATLAB programmer, so I am unable to provide feedback on its use.

---

## Referee Comment (RC2)

**Report for the manuscript**

**"A global monthly field of seawater pH over 3 decades: a machine learning approach" by G. Zhong et al.**

The manuscript presents an application of machine learning techniques to reconstruct global fields of seawater pH covering the years 1992 to 2020 at 1° and monthly resolutions. This research complements to the exiting research studies on generating pH values at the surface ocean by providing global maps of seawater pH for depth levels up to 2000 m, essential for understanding ocean acidification and its impacts on ecosystems in the ocean interior.

**General comments:**

Machine learning offers flexible frameworks to link in situ observations of marine carbonate system variables with relevant environmental conditions. The novelty of this research lies on the idea of mapping on direct pH data from the surface down to different depth layers instead of computing through the carbonate system speciation given prior parameters as pCO2, DIC, …. The authors are also able to integrate various data sources, including in-situ measurements and satellite-based datasets as input of pH estimates.

While the manuscript presents the effectiveness of machine learning models (SOM, Stepwise FFNN, FFNN), there is a crucial need for greater transparency regarding model architecture, selection process of predictors, hyperparameter tuning, model accuracy and uncertainty quantification. Including these details would enhance the reliability of the proposed product and interpretability of the study.

1. **Methodologies:**

   Even interpolation with direct pH data may be feasible but model accuracy remains limited due to the modest amount and data coverage of GLODAP pH used for training and validation, that may be much more problematic at the deep sea. More statistics on GLODAP data used for FFNN traning and validation are needed for a comprehensive

evaluation on the model efficiency. It would be worthy to test the proposed method by subsampling with the same GLODAP tracks on the exiting products that offer global maps of pH at some levels depths (see for instance: https://doi.org/10.48670/moi-00015).

The use of all the three ML algorithms seems redundant and would gain uncertainty for pH mappings (particularly with limited number of training data). FFNN itself would be skillful enough to interpolate pH without SOM for regional clustering and stepwise FFNN for predictor selection (see for instance: Broullón et al., 2019; 2020; Chau et al., 2022; 2024). Results derived from SOM shown in this study (e.g. Figures 1 and 8) does do not reflect the sharing patterns of mechanisms in CO2 uptake and storage, e.g., at temperate (equatorial) zones between different oceanic basins. pH estimation based on clustered biomes creates discontinuity at the regional boundary. Clustering and predictor selection are probably available in FFNN training phase as it automatically weights and select input neurons to get optimal output. In addition, the use of lat, lon, depth as predictors potentially locallizes training data and would allow to interpolate pH at non-observed locations. Results after Stepwise FFNN are not accurate enough (see comment for Table 2 in Specific Comments).

Please consider to provide more details on the data preprocessing steps, including how different data sources were transformed and harmonized, as well as elaborate on (1) the choice of many predictors (e.g., why temperature anomaly was used instead of the standard ones; why both number of months and Year Month are needed,...), (2) machine learning algorithms used, and (3) the rationale behind selecting them over other potential models.

2. **Model robustness and Uncertainty Analysis**:

Although the authors have shown the validation against independent datasets from GLODAP, other ML-based methods, and some time series of direct measurements (e.g., HOT, ESTOC, BATS), the examination limited almost for the surface ocean. Evaluation and analysis of model accuracy at seawater depths would provide a clearer understanding of the model performance. Some timeseries stations have offered pH data below the surface (e.g., HOT,

ESTOC, BATS), other sources for data evaluation can be found in Sutton et al., (2019), Lange et al., (2024).

The manuscript would benefit from a more comprehensive uncertainty analysis. It is not convinced that adding the source of RMSE from FFNN [H+] is essential for the quantification of pH uncertainty as the author ultimately produce 4D pH fields from GLODAP pH data (not [H+]. Presenting a comparison between RMSE of pH derived from direct GLODAP pH and from [H+] in the supplementary is more appropriate. Instead, it would be worthy to consider predictors' uncertainty in the the total uncertainty of reconstructed pH.

**Specific Comments:**

Lines 44-45: The two studies report fast decrease in pH in the ocean interior. The authors should reword this sentences.

Lines 45-46: Can be rephrased (for instance: "there remains a need to enhance our understanding of global ocean acidification rates across varying depths.")

Line 49: " the global mapping of" to "global reconstructions of".

Lines 50-51: Gregor and Gruber, (2021) and Chau et al., (2024) have published full datasets of many carbonate variables (pH, DIC, Alkaninity included).

Lines 55-56: "The construct pH product"; this term is not a standard scientific or technical term. Maybe replace with "the proposed product" or "the recontruced pH data".

Line 61: (Lauvset et al., 2022) → update refs for GLODAPv2.2023 and cite right after 2023 version instead.

Line 62: "in-situ temperature"; Reviewer do not see any specific role of temperature in pH mappings throughout the manuscript.

Line 63: Table 1 show most of predictors' products with no depth levels, it is not clear how to map pH with constant values of predictors over depths!

Lines 66-70: many predictors have been used but there are no hints (citations) showing why they should be included in model fitting.

Line 80: "such as the ocean currents product" → should be at the beginning of the sentence.

Table 1: please mention temporal and vertical resolutions of predictors' products used; comments on how to derive transformation for Date (Year, Month,...).

Line 116: "Therefor".

Figure 1: consider to not use SOM (see in General comments).

Lines 153-154: "To mitigate the influence of the FFNN's initial state on predicting values, multiple networks with the same structure but different initial states were trained and their results were averaged"; standard deviation from output averaging should be reported.

Tables 2, 3: consider to not use Stepwise FFNN to select predictors. FFNN itself would choose which inputs benefit model training. Results from Stepwise FFNN are not align with marine $CO_2$ system features and driving mechanisms. For instance, temperature is one of the key factors modulating $CO_2$ absorption over the Arctic and subpolar regions (thus performs impact on pH); however, this predictor was not chosen after Stepwise FFNN. In addition, please clarify the use of both temperature (salinity,...) and their anomalies that are redundant information and may challenge FFNN training!!!

Line 174: RMSE; the metric for validation is not consistent with training (MAE, Line 135).

Line 183: Please clarify how to define H+ here.

Lines 191-206: It's worthy to rework on this section: It is not ease to interpret the uncertainty quantified with pH0 and sigma (see also in General Comments for details).

Lines 212-214: "A better performance of the FFNN was found in the intermediate layer, with testing samples more concentrated on the y=x line. The RMSE in the mixed layer is 0.034, higher than 0.026 in the intermediate layer." Reviewer would expect to see reverse results: there exist very few pH data and predictors' information which are able to support pH estimation in the deep sea than the shallower layer !!! Reconstruction errors (Uncertainty) would much higher there than the surface.

Line 219: remove "predicting" here and elsewhere, FFNN pH is informative enough.

Lines 224-225: Any clarrifications to have errors lower in the deepsea than the surface. I would appreciate of any clarification.

Line 230: "predicting error" is not correct. Please use "prediction error" or "reconstruction error" instead.

Lines 237-238: "The RMSE in the early years was relatively higher than in recent years,

while the number of GLODAP measurements increased with the years (Figure 5c)". Adding curves for number of GLODAP pH in each suplot will help to evidence the statements for Fig 5.

Lines 249 - : Why do the results show for Stepwise FFNN while final recontruction is done with FFNN?

Lines 251 - 254:

"The surface seawater pH of our Stepwise FFNN product decreased by 0.0017 ± 0.0007 yr-1 on average during the past three decades at the BAT station, close to the -0.0018 ± 0.0001 yr-1 of BAT time series observations in the same period (Bates et al., 2020). At the ESTOC station, the Stepwise FFNNN product and time series observations were also well consistent, with the RMSE of only 0.009 and a similar long-term trend (Chau et al., 2022)."

these quotations are not correctly mentioned in refs (Bates et al., 2020; Chau et al., 2022). The pH decreasing rate was about 0.0019 ± 0.0001 per yr over the period 1983-2020 in the former study. Please clarify that the authors have used the data to compute the trends by themselves. Furthermore, Chau et al., (2022) do not provide long-term trend estimates of pH but Chau et al,. (2024).

Line 255: remove "only" in this parapraph and elsewhere as 0.01 in pH is indeed large correspondingly to a difference in 26% of H+ (acidity level)".

Table 4: report uncertainty estimates for all other products.

**References**

Broullón, D., Pérez, F. F., Velo, A., Hoppema, M., Olsen, A., Takahashi, T., Key, R. M., Tanhua, T., González-Dávila, M., Jeansson, E., Kozyr, A., and van Heuven, S. M. A. C.: A global monthly climatology of total alkalinity: a neural network approach, Earth Syst. Sci. Data, 11, 1109–1127, https://doi.org/10.5194/essd-111109-2019, 2019.

Broullón, D., Pérez, F. F., Velo, A., Hoppema, M., Olsen, A., Takahashi, T., Key, R. M., Tanhua, T., Santana-Casiano, J. M., and Kozyr, A.: A global monthly climatology of oceanic total dissolved inorganic carbon: a neural network approach, Earth Syst. Sci. Data, 12, 1725–1743, https://doi.org/10.5194/essd-12-1725-2020, 2020.

Chau, T. T. T., Gehlen, M., and Chevallier, F.: A seamless ensemble-based reconstruction of surface ocean pCO2 and airsea CO2 fluxes over the global coastal and open oceans, Biogeosciences, 19, 1087–1109, https://doi.org/10.5194/bg-19-10872022, 2022b.

Chau T T T, Gehlen M, Metzl N, Chevallier F (2024). CMEMS-LSCE: A global 0.25-degree, monthly reconstruction of the surface ocean carbonate system, Earth System Data, 16, 121–160, https://doi.org/10.5194/essd-16-121-2024.

Lange, N., Fiedler, B., Álvarez, M., Benoit-Cattin, A., Benway, H., Buttigieg, P. L., Coppola, L., Currie, K., Flecha, S., Gerlach, D. S., Honda, M., Huertas, I. E., Lauvset, S. K., Muller-Karger, F., Körtzinger, A., O'Brien, K. M., Ólafsdóttir, S. R., Pacheco, F. C., Rueda-Roa, D., Skjelvan, I., Wakita, M., White, A., and Tanhua, T.: Synthesis Product for Ocean Time Series (SPOTS) – a ship-based biogeochemical pilot, Earth Syst. Sci. Data, 16, 1901–1931, https://doi.org/10.5194/essd-16-1901-2024, 2024.

Sutton, A. J., Feely, R. A., Maenner-Jones, S., Musielwicz, S., Osborne, J., Dietrich, C., Monacci, N., Cross, J., Bott, R., Kozyr, A., Andersson, A. J., Bates, N. R., Cai, W.-J., Cronin, M. F., De Carlo, E. H., Hales, B., Howden, S. D., Lee, C. M., Manzello, D. P., McPhaden, M. J., Meléndez, M., Mickett, J. B., Newton, J. A., Noakes, S. E., Noh, J. H., Olafsdottir, S. R., Salisbury, J. E., Send, U., Trull, T. W., Vandemark, D. C., and Weller, R. A.: Autonomous seawater pCO2 and pH time series from 40 surface buoys and the emergence of anthropogenic trends, Earth Syst. Sci. Data, 11, 421–439, https://doi.org/10.5194/essd11-421-2019, 2019.

---

## Author Comment (AC1)

**Reviewer Comment #1**

**Context and general comment :**

The continuous absorption of anthropogenic $CO_2$ by the ocean leads to ocean acidification, which threatens marine ecosystems. While the acidification rate has been extensively documented at the surface, data for deeper waters remain limited. Zhong et al. address this gap by presenting a comprehensive, monthly, four-dimensional, 1°×1° gridded global seawater pH dataset, covering the years 1992 to 2020 and depths from the surface to 2000 meters.

This dataset was developed using machine learning algorithms trained on pH observations from the Global Ocean Data Analysis Project (GLODAP). The methodology employed is a three-step process: 1) self-organizing map neural network for bioregionalization, 2) a stepwise algorithm for predictors selection, and 3) feed-forward neural networks (FFNN) for non-linear regression. The resulting pH product is a valuable resource for studying subsurface ocean acidification and for validating or initializing biogeochemical models. The product is made publicly available through the Marine Science Data Center of the Chinese Academy of Sciences.

Overall, the article is well-written and the figures are clearly presented.

Despite the significance of this new 3D pH product for the scientific community, the article has some notable shortcomings. There is a lack of details in the methodology section, which makes it challenging to fully evaluate the robustness of the method and comprehend the implications involved.

Response: Thank you very much for agreeing that our data has significant value. The concern about the lack of details in the methodology section is important and helpful for us to improve the manuscript quality. We have revised the manuscript according to the comments, which may provide a clearer understanding of the methods used in this work. The detailed changes were listed in the response to specific comments.

**Specific Comments:**

**Title:**

It may be valuable to the reader to add the information that the estimations are depth-resolved, resulting in a 3D product, which is the principal novelty of this methodology.

Response: Thanks for the suggestion. The title has been changed to "A global monthly 3D-field of seawater pH over 3 decades: a machine learning approach".

**Abstract:**

*- Lines 15-17:* "Here, we present a monthly four-dimensional 1°×1° gridded product of global seawater pH, derived from a machine learning algorithm trained on pH observations at total scale and in-situ temperature from the Global Ocean Data Analysis Project (GLODAP).": The role of temperature in the methodology is unclear. Even after reading the entire paper, the specific role of temperature compared to other inputs remains ambiguous.

Response: Temperature is an important factor affecting seawater pH, and was used as a pH predictor in almost all regions in this work. In many regions temperature is considered one of the most important predictors and ranks very high in the predictor list in Tables 2 and 3, as the sort order represents the predictor's capacity to reduce predicting errors. Except the relative importance, the role of temperature is the same as other predictors listed in Tables 2 and 3. All selected predictors were treated with same process and input into the neural networks.

The reason for mentioning temperature here is not to show its important role, but to distinguish with the product standardized to a temperature of 25°C. For better clarity, the original text has been modified as the following:

"Here, we present a monthly four-dimensional 1°×1° gridded product of global seawater pH at total scale and in-situ temperature, derived from a machine learning algorithm trained on pH observations from the Global Ocean Data Analysis Project (GLODAP)."

*- Line 18:* I suggest rephrasing the method description for clarity. Consider stating: "A three-step machine learning-based algorithm was used..."

Response: Thanks for the suggestion. The original text has been modified as the following:

"A three-step machine learning-based algorithm was used to construct the pH product, incorporating region division by the self-organizing map neural network, predictor selection by the stepwise regression algorithm that adds and removes variables from network inputs based on their contribution to reducing predicting errors, and non-linear relationship regression by feed-forward neural networks (FFNN)."

- *Line 19:* The term "stepwise" may not be clear to the readers. Consider elaborating or using a more descriptive term.

Response: Thanks for the suggestion. The term "stepwise" has been changed to "the stepwise regression algorithm that adds and removes variables from network inputs based on their contribution to reducing predicting errors" for better clarity.

**Introduction:**

- The introduction appears to be missing some crucial references. For example, it would be beneficial to acknowledge that the methodology is inspired by the work of Landschützer et al. (2014) and references following; i.e. a SOM-FNN approach. Additionally, it is important to mention that this SOM-FNN approach has already been applied to the 3D reconstruction of DIC by Keppler et al., 2020 (https://agupubs.onlinelibrary.wiley.com/doi/10.1029/2020GB006571). These references should be cited to provide a more comprehensive background.

Response: Thanks for correcting me. Landschützer is a scientist whom I highly respect, and I have learned a lot of knowledge from his work. The missing references have been added in the introduction and the text has been modified as the following:

"Recent applications of machine learning methods in the global mapping of marine carbonate system variables have facilitated global-scale research on the acidification and carbon cycle, including the single/ensemble-based FFNN method and the SOM-FFNN method for mapping surface ocean partial pressure of $CO_2$ ($pCO_2$, Landschützer et al., 2014; Chau et al., 2022; Zhong et al., 2022), dissolved inorganic carbon (DIC, Broullón et al., 2020; Keppler et al., 2020), and alkalinity (Broullón et al., 2019; Gregor and Gruber, 2021). Additionally, the 3D-field mapping of DIC was also conducted using the SOM-FFNN method, which produced monthly climatological data but lacked interannual variability (Keppler et al., 2020). These methods have inspired our methodology for constructing the global gridded seawater pH dataset. Until now, only surface ocean gridded pH products are available in acidification research, including the 1° JMA product (Iida et al., 2021), the 1° OceanSODA-ETHZ product (Gregor and Gruber, 2021), the 0.25° remote-sensing-based product (Jiang et al., 2022), and the 0.25° CMEMS-LSCE product (Chau et al., 2024), which were derived from mapping $pCO_2$, DIC, or alkalinity using machine learning algorithms and subsequently calculating pH with the CO2SYS program (Lewis and Wallace, 1998)."

- *Line 52 :* where existing pH surface products are listed, there is a lack of references and details. It is crucial to include comprehensive citations of existing products (e.g. LSCE-FFNN), especially those used for comparison in Table 4 (+ the missing ones, see my comment below).

Response: Thanks for the suggestion. The citation of Chau et al., 2024 for the CMEMS-LSCE product and Jiang et al., 2022 for their remote sensing product have been added. The product names, spatial resolutions, and details in method has been added as the following:

"Until now, only surface ocean gridded pH products are available in acidification research, including the 1° JMA product (Iida et al., 2021), the 1° OceanSODA-ETHZ product (Gregor and Gruber, 2021), the 0.25° remote-sensing-based product (Jiang et al., 2022), and the 0.25° CMEMS-LSCE product (Chau et al., 2024), which were derived from mapping $pCO_2$, DIC, or alkalinity using machine learning algorithms and subsequently calculating pH with the CO2SYS program (Lewis and Wallace, 1998)."

- The paper does not explain why the product spans the period 1992-2020. It would be helpful to provide a rationale for this timeframe and discuss why it does not cover a longer period, both in the past and up to the present (i.e., year-1).

Response: Thanks for the suggestion. The product starts from 1992 because the SSH, MLD, and W velocity of ocean currents used as pH predictors from the ECCO2 cube92 product also start from 1992. Data after 2020 is unavailable because our surface ocean $pCO_2$ product has not been updated beyond 2020. The pH data will be updated to cover the period 1992-2023 after the update of our $pCO_2$ product and the NOAA Greenhouse Gas Marine Boundary Layer Reference product (currently updated only until 2022). The explanation for the covered period has been added at the end of the section "2.3 pH product construction" as the following:

"The pH data earlier than 1992 is unavailable because the predictors used from ECCO2 cube92 product (Menemenlis et al., 2008) also start from 1992. Data after 2020 is limited by the coverage of used surface ocean $pCO_2$ product and will be updated in future works."

- *Line 55:* The reference to GLODAP by Lauvset et al. (2022) refers to GLODAPv2.2022 and should be cited as such throughout the article (instead of 'GLODAP'). Additionally, if the authors re-run their model during the review process, it is suggested to use the latest version of GLODAP, i.e., GLODAPv2.2023. The

updated        reference        can        be        found        here:
https://essd.copernicus.org/articles/16/2047/2024/essd-16-2047-2024-discussion.html

Response: Thanks for the suggestion. The latest version of GLODAP has been used, and the citation has been updated to Lauvset et al., 2023 and added in all places that GLODAP was mentioned.

**Methods:**

*Line 60:* It is unclear how temperature is used in the methodology. Additional explanation is needed to clarify its specific role and contribution.

Response: Here we mentioned temperature because the pH reconstructed in this work is at in-situ temperature, not at the commonly used 25°C. Additionally, temperature was used as one of pH predictors and played the same role with other variables technically, as shown in Formula 1 ($pH = f(\text{Predictors}_1, \text{Predictors}_2, \dots, \text{Predictors}_N)$). Here in-situ temperature was mentioned to avoid the product being considered a 25°C product, as some researchers may convert the pH measurements or pH products to a normalized temperature of 25°C.

*Lines 69-73:* The inclusion of other indices, such as the Northern Oscillation Index, should be considered.

Response: Thanks for the suggestion. We believe that parameters like the Northern Oscillation Index have smaller impact on pH than currently used indices, and are not considered here. However, we will evaluate the impact of the Northern Oscillation Index on pH in future work. These indices will be added in the future works to be tested if they can contribute to reduction of pH predicting errors.

*Lines 71-73:* The LSCE-FFNN product provides total alkalinity and DIC data monthly from 1985 to year-1: data available from the Copernicus Marine Service (https://data.marine.copernicus.eu/product/MULTIOBS_GLO_BIO_CARBON_SUR FACE_REP_015_008/description). If authors refer to 3D products, then it has to be clearly mentioned. For 3D estimations, DIC is available monthly from 2004 to 2019 from MOBO-DIC (Keppler et al., 2020).

Response: Thanks for the suggestion. The description was about 3D products of DIC and alkalinity, and has been corrected as the following:

"However, 3D field products with sufficient time and spatial coverage are currently not available for these two variables, so climatological 3D products were used for better pH spatial distribution."

*Section 2.2 :*

*- Lines 103-105:* The rationale for using these specific parameters to define bioregions needs clarification, especially if they are not significant in the stepwise algorithm for determining important parameters in relation to pH.

Response: Thanks for the suggestion. The parameters used to define bioregions are related to physical and biological processes affecting pH, with most being selected as predictors in many bioregions, except for chlorophyll concentration. However, this does not imply that chlorophyll concentration is unrelated to pH changes. Chlorophyll concentration is highly related to photosynthesis, which affects pH by influencing $p$CO$_2$. Its role as a pH predictor is replaced by $p$CO$_2$. Currently, the uncertainty of $p$CO$_2$ products is higher than that of satellite remote sensing chlorophyll products, and using different $p$CO$_2$ products may affect the dividing of bioregions. Although the current bioregions are not perfect, we have set a small number of bioregions with broad coverage to reduce the impact of potential inaccuracies in bioregions division. In future work, we will further improve the parameters used to define bioregions based on the currently selected predictors.

*- Line 106 :* The criteria and process for merging provinces with fewer than ten connected grids or less than 100 GLODAP pH measurements should be rephrased and/or detailed because it is unclear.

Response: Thanks for the suggestion. The SOM may accidently generate three types of small "island" provinces: provinces consisting of many disconnected grids across different regions, provinces covering very small areas with fewer than 10 connected grids, and provinces with an insufficient number of GLODAP pH measurements. These provinces are not helpful in the pH product construction, as the pH predicting errors tend to be notably higher due to the extremely few training samples for FFNNs. A similar process can be also found in previous SOM-based research, such as Landschützer et al. (2014), which removed small "island" provinces with a surface area smaller than 10 connected grid cells. The origin text has been modified as the following: "Subsequently, the small "island" provinces with fewer than ten connected grids or covered by fewer than 100 GLODAP pH measurements were merged with the nearest neighboring provinces, as the pH predicting errors tend to be notably higher due to the extremely few training samples in the non-linear relationship fitting by networks."

*- Line 107 :* The need for manual subdivision of provinces separated by continents requires further explanation. Why not using same bioregion even if it is not in the same

ocean, it is possible that the underlying processes are equivalent and so that the FFNN will be performant in both basin?

Response: Thanks for the suggestion. The physical process may also affect seawater pH distribution and are different between basins separated by continents, such as surface ocean currents. However, these physical mixing and transport processes are actually separated by lands, so we manually divided provinces separated by continents.

- *Lines 112-114 :* The sentence on the division of ocean areas into different layers also requires further details on which drivers are important for each layer as it is stated that drivers differ depending on the layers. Moreover, following this statement, why using the same bioregions for deeper layers?

Response: In the mixed layer, seawater pH is notably influenced by the seasonal cycle of environmental conditions and $CO_2$ exchange between the surface ocean and atmosphere. In the intermediate layer, seawater pH experiences much weaker seasonal changes and is largely affected by the biological drawdown of organic matter. Such differences in pH drivers can also be observed in the selected predictors in Tables 2 and 3, with the most important predictor in many provinces being surface ocean $p$CO$_2$ in the mixed layer and phosphate concentration in the intermediate layer.

The variables used for division of regions are fully available in the surface ocean, but some variables lack information at different depth, making it difficult to divide regions separately in the deep ocean. Additionally, to maintain consistency in geographic regions between the two vertical layers with existing vertical seawater mixing, it is unnecessary to divide regions separately in the mixed and intermediate layers as they are not completely separate areas. Therefore, we applied the surface biogeochemical provinces to the deeper ocean as well.

*Section 2.3 and Table 1 :*

- The choice of a single-layer FFNN instead of a multi-layer network should be justified. Has this been tested ?

Response: The FFNN with single hidden layer has a smaller scale with faster training and calculating speed, and is also convenient for adjusting the number of neurons. We also compared different neural network structures. With a similar number of neurons, the impact of different structures on error is relatively small. By increasing the number of neurons, a fitting capability comparable to that of multi-layer neural networks can be achieved. Additionally, in previous studies reconstructing surface ocean $p$CO$_2$ gridded data, we also used single-layer neural networks and verified their sufficient

fitting capability to reconstruct global ocean gridded product of carbonate system variables.

- The use of sin(Lat) as a predictor is questionable since latitude is not circular.

Response: This normalization method was inspired from previous research, such as Denvil-Sommer, A., et al. (2019), where they also normalized latitude and longitude to radians using sine and cosine transformations. Also, we have corrected the description name in Table 1 to "Sine of (latitude · $\pi$/180°)", "Sine of (longitude · $\pi$/180°)", and "Cosine of (longitude · $\pi$/180°)". As we used the "sind" and "cosd" function (sind(latitude) equals sin(latitude · $\pi$/180°)) in MATLAB, the original description was misleading and has been corrected.

Denvil-Sommer, A., et al. (2019). "LSCE-FFNN-v1: a two-step neural network model for the reconstruction of surface ocean $p$CO$_2$ over the global ocean." Geoscientific Model Development 12(5): 2091-2105.

- Clarify how depth is used as a predictor and whether it corresponds to the depth of retrieval of the output or if the FFNN estimates X values for X depth levels.

Response: Thanks for the suggestion. Depth was used in the same way as latitude or time-related variables in Table 1. The sample depths of GLODAP measurements were input into FFNNs during the training process, and the depths of 41 depth layers defined as target output layers were input into FFNNs during the interpolation process to generate a product covering 0-2000m. The description has been added in the 2.1 section as the following:

"Temporal and spatial sample information, including latitude, longitude, depth and sample time, was also used as supplementary variables. Latitude and longitude were normalized to radians using sine and cosine transformations, to present connected sample position information. The spatial sample position and time information of GLODAP measurements were input in the training of FFNNs, and the spatial position and time of defined 1° and monthly product grids were input into FFNNs during the interpolation process to output a gridded product."

- The choice of the ECCO2cube92 model should be discussed and better supported by citations in the text.

Response: Thanks for the suggestion. We used the ECCO2 cube92 product due to its wide temporal and spatial coverage, which starts from 1992 and is continuously

updating to the present. Moreover, the MLD and SSH data from ECCO2 were also used in previous research on mapping of ocean carbonate system variables and proved the reliability, for example, Landschützer, et al. (2014) and Chau, et al. (2024). A description has been added in "2.1 Data sources and processing" section as the following:

"Products used for variables listed in Table 1 was chose due to their sufficient temporal and spatial coverage and the application in previous research on mapping carbonate system variables. For example, the ECCO2 MLD product has been used in reconstruction of the CMEMS-LSCE surface ocean carbonate system variables product (Chau, et al., 2024) and the MPI-SOM-FFN pCO2 product (Landschützer et al., 2014)."

Landschützer, et al. (2014). Recent variability of the global ocean carbon sink. Global Biogeochemical Cycles 28(9): 927-949.

Chau, et al. (2024). CMEMS-LSCE: a global, 0.25°, monthly reconstruction of the surface ocean carbonate system. Earth System Science Data 16(1): 121-160.

- MEI should be defined as the Multivariate ENSO Index.

Response: Thanks for the suggestion. The name of MEI has been corrected.

- Adding a column to Table 1 to indicate which process each variable is associated with would be informative.

Response: Thanks for the suggestion. The related processes have been added in Table 1 as the following:

**Table 1. Data products used as pH predictors.**

[revised manuscript text omitted]

- Consider using merged satellite ocean color data products like OC-CCI or GlobColour for longer time series would help for future usability and sustainability.

Response: Thanks for the suggestion. These products will be used when we update our product.

- Provide details on how the most informative parameters were chosen and how hyperparameters (architecture, number of neurons) were handled in this stepwise process.

Response: Thanks for the suggestion. We have revised the predictor selection section, and added the specific procedure of stepwise FFNN algorithm in Figure 3 as the following:

[revised manuscript text omitted]

- Clarify how co-correlation among selected predictors was removed in the stepwise FFNN selection procedure.

Response: Since the variables with co-correlations provide similar information, the predicting error using two co-correlated predictors will be higher than that using two predictors related to different ocean processes. Therefore, the variable correlated to existing predictors tends to fail to compete with other variables in the predictor selection. Moreover, whenever after a new predictor is identified, the stepwise FFNN algorithm also tests whether the predicting error will decrease when sequentially removing each

determined predictor. If a certain predictor is highly correlated with existing predictors, this predictor will be generally removed for to reduce in predicting errors.

- The sentence regarding additional FFNNs trained with predictors in Table S1 for polar areas and periods before August 2002 needs clarification.

Response: The sentence has been modified as the following:

"Since the satellite remote sensing products used in this work lack data during the period before August 2002 and in certain polar areas during winter, the FFNN generated missing values in these grids when remote sensing products were used as predictors. To address these missing values, we selected additional groups of predictors after removing remote sensing products (Table S1), and then trained additional FFNNs to predict pH in grids with missing values. This procedure was the same as the reconstruction process in the intermediate layer, in which the remote sensing products were also not used."

- Discuss Tables 2 and 3 scientifically in the Results section to highlight important processes driving pH variability.

Response: Thanks for the suggestion. We have added a discussion about processes driving pH variability in the end of section 3.2.1 Spatial pH distribution as the following:

"Based on the pH predictors selected by the Stepwise FFNN algorithm, differences in processes driving pH variability were identified between the mixed layer and intermediate layer in most provinces. In the mixed layer, surface ocean $pCO_2$ was identified as the most informative predictor in many provinces, followed by temperature and nutrient concentration. This suggests that the $CO_2$ exchange between surface ocean and atmosphere is the primary driver of pH variability, followed by biological $CO_2$ utilization and seasonal changes in seawater temperature. In contrast, phosphate was identified as the most informative predictor in the intermediate layer, followed by temperature and depth. This suggests that the primary process driving pH variability is the remineralization of organic matter, converting organic carbon into inorganic forms and also releasing nitrogen and phosphorus. Given the notably smaller seasonal temperature changes in the intermediate layer compared to the mixed layer, the selection of temperature as an important pH predictor may indicate a notable influence of ocean warming on seawater pH variability. Additionally, depth was also selected as an important predictor in the intermediate layer. The observed pattern of seawater pH decreasing with increasing depth in most provinces, as suggested by the constructed pH product, may be the main reason."

- Figure 3 is extremely difficult to understand and should be clarified or redesigned.

Response: Thanks for the suggestion. We have revised Figure 3 as the following:

[Figure]

Figure 3: The procedure of pH product construction. (1) Stepwise FFNN: the algorithm for selecting predictors (Zhong et al., 2022); (2) FFNN: fitting the non-linear relationship between seawater pH and its predictors. Collected Environmental variables: collected products listed in Table 1. pH predictors: the selected most informative variables listed in Tables 2 and 3. Remote sensing products: variables from Chlorophyll to Total backscattering in Table 1. Mixed layer: from 0 m to mixed layer depth; intermediate layer: from mixed layer depth to 2000 m.

More generally, the section 2.3 is currently unclear and needs to be rewritten with more detailed explanations.

Response: Thanks for the suggestion. We have revised the section 2.3 and added more details about the method. The specific changes can be found in above response.

*Section 2.4:*

*Line 191:* The paragraph is unclear. The statement, "Therefore, the uncertainty of our pH product was directly estimated from the FFNN pH predicting errors, instead of synthesizing the inherent uncertainty of each used predictor product," needs further clarification. How was this done?

Response: As described in equation (2), the uncertainty was estimated from local pH value and pH predicting error in the corresponding province. For the uncertainty in certain grid, we first convert pH predicting error in the corresponding province into difference of $[H^+]$, by logarithm transfer of predicted and GLODAP measured pH and then calculating RMSE. Subsequently, the RMSE of [H+] was transferred to pH uncertainty based on the local pH value.

$$\sigma = -\log_{10}(10^{-pH_0} - RMSE_{[H^+]}) - pH_0$$

where $RMSE_{[H^+]}$ was the RMSE of $[H^+]$ converted from FFNN pH predicting error in each vertical layer and in each biogeochemical province. $pH_0$ was the local predicted pH value in the grid that uncertainty was estimated. Due to missing inherent uncertainty of particular predictor product, estimating uncertainty from inherent uncertainty of used predictor products was unfeasible.

*Section 3.1:*

*- Line 214:* This interpretation might be overstated. The broader value range likely contributes to a better model fit, and pH values exhibit less variability at depth.

Response: Thanks for the suggestion. The sentence has been modified as the following:

"The minor difference between the predicting value and the pH measurements and the R2 of 0.97 in the intermediate layer may be caused by less pH variability at depth and better model fit with broader pH value range."

*- Figure 4:* Authors might add the slopes of the linear regression to the statistics.

Response: The slopes and linear regression lines have been added.

*- Line 235:* The impact of the Oxygen Minimum Zone (OMZ) on the product should be discussed more in details.

Response: Although dissolved oxygen was considered the most informative predictor in the Indian Ocean, statistical differences in pH predicting errors were not observed between the OMZ and other areas. We have added a description about the impact of the OMZ on the quality of pH product. The impact of the OMZ on the pH variability will

be investigated in future work, as much more efforts are needed and the currently used dissolved oxygen product is only monthly climatological.

- *Figure 5:* This figure requires re-arrangement. The map should be larger, and pH differences against depth should be plotted with depth as the y-axis, as is more common for reading profiles. Additionally, including seasonal variability for each major basin along with yearly variability would be beneficial.

Response: Thanks for the suggestion. We have re-arrangement Figure 5. It is difficult to compare seasonal variability for each major basin, as the seasonal variability of the north and south hemispheres in the Pacific and Atlantic Ocean cancel each other out.

[Figure]

Figure 5: Distribution of RMSE between FFNN predicted pH values and GLODAP pH measurements. a): global spatial distribution of RMSE between FFNN predicted pH and GLODAP pH measurements at 0-2000 m; b): basin average RMSE at different depth; c): temporal distribution of global RMSE; d): Statistical distribution of pH difference between predicted pH values and GLODAP pH measurements in each basin.

- In the validation section, it would be valuable to compare the global scale trend with the Copernicus Marine Service data: https://marine.copernicus.eu/access-data/ocean-monitoring-indicators/globalocean-acidification-mean-sea-water-ph-time-series.

Moreover, it would be interesting to add comparison against qualified pH data from BGC-Argo dataset.

Response: Thanks for the suggestion. Comparison of global scale trend has been added in Table 4. The BGC ARGO pH data qualified by IMOS has been added in the validation section. Different from the validation results based on the GLODAP dataset, the RMSE between FFNN pH and BGC ARGO pH data is higher in the deep ocean. Only the bias between FFNN pH and BGC ARGO pH data tends to increase with depth in most basins. In contrast, greater biases between FFNN pH and GLODAP pH occur mainly in the surface layer. Especially in the Southern Ocean, the bias between FFNN pH and GLODAP pH is nearly zero below 1000 m, notably lower than biases between FFNN pH and BGC ARGO pH data ranging from 0.053 to 0.076. This may be primarily attributed to the discrepancies between GLODAP dataset and the BGC ARGO dataset in the deep ocean, as our product was based on GLODAP dataset and small biases with GLODAP pH were observed in the deep ocean.

[Figure]

**Figure 8. Difference between FFNN pH and BGC ARGO floats pH.** a) comparison between FFNN pH and BGC ARGO floats pH in the mixed layer; b) comparison between FFNN pH and BGC ARGO floats pH in the intermediate layer; c) Statistical distribution of pH difference (FFNN pH minus BGC ARGO floats pH) at different depth levels. FFNN pH: pH data reconstructed in this work; BGC ARGO floats pH: pH data from IMOS Biogeochemical ARGO floats core data collection (IMOS 2002-2020, 2021).

**Table 5. pH bias by area and depth computed with BGC ARGO and GLODAP dataset.**

| Area | | 0-50 m | 50-200 m | 200-500 m | 500-1000 m | 1000-1500 m | 1500-2000 m |
|---|---|---|---|---|---|---|---|
| Arctic | BGC ARGO | -0.006 | 0.021 | 0.014 | 0.026 | 0.037 | 0.029 |
| | GLODAP | -0.005 | -0.003 | 0.001 | 0.000 | 0.002 | 0.003 |
| Pacific | BGC ARGO | 0.012 | 0.011 | -0.008 | -0.015 | 0.008 | 0.000 |
| | GLODAP | -0.001 | -0.001 | 0.000 | 0.000 | 0.000 | -0.001 |
| Atlantic | BGC ARGO | 0.016 | 0.017 | 0.010 | -0.024 | 0.029 | 0.068 |
| | GLODAP | 0.000 | 0.000 | -0.001 | -0.001 | 0.000 | 0.000 |
| Indian | BGC ARGO | 0.024 | 0.026 | 0.014 | -0.056 | 0.002 | 0.059 |
| | GLODAP | -0.006 | -0.001 | -0.003 | -0.004 | -0.004 | -0.001 |
| Southern | BGC ARGO | 0.011 | 0.002 | 0.002 | 0.027 | 0.053 | 0.076 |
| | GLODAP | 0.004 | 0.001 | 0.001 | 0.000 | 0.000 | 0.000 |
| Global | BGC ARGO | 0.011 | 0.005 | 0.001 | 0.021 | 0.046 | 0.066 |
| | GLODAP | -0.001 | 0.000 | 0.000 | -0.001 | -0.001 | 0.000 |

- *Figure 6 + text:* Comparing to other available pH time series would be interesting. These are listed in the recent ESSD paper by Lange et al. (2024): https://essd.copernicus.org/articles/16/1901/2024/. For instance, the Mediterranean Sea, where data from GLODAP V2 are very scarce, could be validated against the Dyfamed pH time series.

Response: Thanks for the suggestion. We only compared with three station due to their sufficient availability in temporal coverage. The collection by Lange et al. (2024) includes many stations and are helpful for further evaluating our product, but the pH data of particular stations are only available for several years. Therefore, we only added the Iceland Sea, the Irminger Sea, and the DYFAMED station data in Table 4 as the following:

**Table 4: Comparison of surface acidification rate with previous product in different time series stations and on a global scale.**

| Stations | Period | Time series observation | Stepwise FFNN (This study) | JMA (Iida et al., 2021) | CMEMS (Chau et al., 2024) | OS-ETHZ (Gregor et al., 2021) | Copernicus (Copernicus Marine Service, 2020) |
|---|---|---|---|---|---|---|---|
| BAT | 1992~2020 | -0.0018 ± 0.0001 | -0.0017 ± 0.0007 | -0.0018 ± 0.0002 | -0.0018 ± 0.0002 | -0.0018 ± 0.0002 | - |
| ESTOC | 1995~2010 | -0.0016 ± 0.0001 | -0.0014 ± 0.0005 | -0.0022 ± 0.0003 | -0.0020 ± 0.0002 | -0.0017 ± 0.0003 | - |
| HOT | 1992~2020 | -0.0018 ± 0.0001 | -0.0018 ± 0.0004 | -0.0020 ± 0.0001 | -0.0021 ± 0.0001 | -0.0019 ± 0.0001 | - |
| Iceland Sea | 1992~2019 | -0.0020 ± 0.0004 | -0.0028 ± 0.0002 | -0.0030± 0.0003 | -0.0015 ± 0.0002 | -0.0020 ± 0.0002 | - |
| Irminger Sea | 1992~2019 | -0.0025 ± 0.0004 | -0.0022 ± 0.0002 | -0.0027 ± 0.0002 | -0.0017 ± 0.0003 | -0.0016 ± 0.0003 | |
| DYFAMED | 1998~2017 | -0.0010 ± 0.0008 | -0.0005 ± 0.0003 | - | -0.0017 ± 0.0003 | -0.0023 ± 0.0004 | |
| Global | 1992~2020 | - | -0.0015 ± 0.0002 | -0.0018 ± 0.0000 | -0.0017 ± 0.0004 | -0.0018 ± 0.0000 | -0.0017 ±0.0002 |

- *Figure 6:* Discuss the extreme values not reconstructed by the FFNN in the text.

Response: The extreme low values not reconstructed by the FFNN are mainly observed at the BAT station near 2010 and at the HOT station near 2000, under the influence of La Niña events. The extreme high values are mainly observed at the HOT station before 2000, under the influence of El Niño events. Differently, the extreme values not reconstructed by the FFNN are less observed at the ESTOC station, where the surface pH did not notably fluctuate during El Niño/La Niña events. It can be inferred that the extreme values not reconstructed by the FFNN may be due to its underestimating of the impact of El Niño/La Niña events on pH of certain temperate areas.

- *Line 254 :* Chau et al. (2022) may not be the best reference, as they are also model (ML)-based.

Response: The reference has been corrected to González-Dávila et al. (2010).

- *Line 261* : Describe in what specific ways the product differs from other products.

Response: The long-term pH trend of our product at the ESTOC station was only -0.0014 yr$^{-1}$, notably slower than the trend from -0.0017 yr$^{-1}$ to -0.0022 yr$^{-1}$ of other gridded products, but is still close to the -0.0016 ± 0.0001 yr$^{-1}$ of real observations.

- *Table 4:* More products could be compared, such as Jiang et al. (2022): Remote Sensing of Global Sea Surface pH Based on Massive Underway Data and Machine Learning (https://doi.org/10.3390/rs14102366). Additionally, some products compared here have not been previously cited in the article (refer to the comment on the introduction). The effect of the different time ranges of the different products on the computation of trends should also be analyzed and discussed.

Response: Thanks for the suggestion. The comparison of surface pH trend in time series stations and on a global ocean scale between different products was based on the same

period, which all contain years before 2004 not covered by product of Jiang et al. (2022). Therefore, the product of Jiang et al. (2022) was not used, but we added the Copernicus Marine Service product for comparison instead, which has a longer time coverage. The citations of products have been added in the introduction. The pH trends from other products for comparison were all re-calculated using data during same periods, and the description has been added in table remarks.

*Section 3.1.2 and Figure 7:* Not sure whether this paragraph and figure are necessary.

Response: These contents has been moved to supplement.

*Section 3.2:*

*- Lines 301-304:* This issue is problematic and should be discussed in more details for the user. Additionally, the significant differences between the GLODAP climatology and this product at 1000 m in the Southern Ocean should be discussed/addressed.

Response: The description has been corrected to focus on the temperate Pacific as the following:

"In the temperate Pacific Ocean, differences in surface pH levels were observed between the west and east in both our product and GLODAP pH climatology, which may be caused by the spread of eastern equatorial seawater with extremely low pH. At the deeper depth of 1000 m, the spatial distribution pattern of FFNN pH product is generally consistent with the GLODAP climatology, despite still some disturbance of bad FFNN performance along the SOM province boundary and the higher FFNN pH in the Southern Ocean. However, the distribution of higher FFNN pH in the region between 35°S and 50°S is consistent with the lower DIC reconstructed by Broullon., et al. (2020)."

The difference in the Southern Ocean may be because the pH observations are sparse and uneven in time and space in the deep Southern Ocean, leading to some pH distribution differences in local areas or depths. However, in the high-latitude regions of the Southern Ocean, our constructed data is also in good agreement with the GLODAP climatology, with an average pH of around 7.9. In the region between 35°S and 50°S, the distribution of higher FFNN pH is consistent with the lower DIC reconstructed by Broullon, et al. (2020).

*- Figure 9:* The longitude of the zonal average should be specified in the caption and/or the text.

Response: The pH values shown at each latitude were averaged from pH values across all longitudes within each major basin. The description has been added in the caption.

*-Section 3.2.2:* The discontinuity problem requires more discussion, both methodologically (explaining why this issue occurs despite the use of the cross-boundary method) and in terms of implications for users. If local uncertainties are available, they should be included in the NetCDFs.

Response: Thanks for the suggestion. The further discussion has been added in the Section 3.2.2. The local uncertainties have been added in the NetCDF files. Here are the added text:

"Although we have used a cross-boundary method to improve the FFNN performance near the SOM and vertical boundary, there are still some discontinuity problems and relatively higher uncertainty. This is because the pH values on two sides of the SOM boundary were reconstructed from two different FFNN models, which were trained with different samples and used different predictors. If one of the FFNN models experienced a worse performance due to insufficient training samples or predictors, the pH values on two sides of the SOM boundary will still differ notably, that is, discontinuity along the boundary. Therefore, the analyze on a regional scale based on pH values near SOM boundaries should be more cautious when using our product."

*Section 5:*

Authors should provide more concrete examples of applications for their product in the Conclusion.

Response: The description of applications has been added in the Section 5 as the following:

"This product serves as a reference for guiding acidification surveys by providing a general understanding of acidification process at different depths on a basin scale and indicating areas with potential fast or slow acidification rates. Additionally, the pH product brings insights into acidification research and can be used to analyze the influence of specific ocean processes on acidification rates and the broader impacts of acidification on a large scale when direct observations are unavailable."

**Typos :**

*- Line 78:* was converted to a 1°×1° resolution by averaging **16** 0.25° grids into one 1° grid

*- Line 84:* **(***

*- Line 116:* Therefor**e**

Response: These typos have been corrected.

**Data:**

I encountered an error when attempting to open the NetCDF file using R. The error message was as follows:

Dans nc_open("/home/user/Data/2012.nc") :

WARNING file /home/user/Data/2012.nc is not compliant netCDF; variable pH is numeric but has a character-type missing value! This is an error! Compensating, but you should fix the file!

Although I didn't receive any warnings when using xarray with Python, this issue should be addressed to ensure compatibility with other tools. Additionally, when opening the dataset with xarray and attempting to plot it using the library's functions, I noticed that the longitude and latitude are reversed (not in the name), and the longitude is plotted on the y-axis. To enhance user-friendliness when using Python tools, it would be beneficial to adjust the format accordingly.

Regarding the availability of MATLAB code, I am not a MATLAB programmer, so I am unable to provide feedback on its use.

Response: This error is caused by the missing value defined as "nan" in the NetCDF file, which can be read by MATLAB but maybe not in other tools. We will replace the missing values to "-999" for all NetCDF files, to ensure compatibility with other tools.

---

## Author Comment (AC2)

**Reviewer Comment #2**

**Report for the manuscript**

**"A global monthly field of seawater pH over 3 decades: a machine learning approach" by**

**G. Zhong et al.**

The manuscript presents an application of machine learning techniques to reconstruct global fields of seawater pH covering the years 1992 to 2020 at 1° and monthly resolutions. This research complements to the exiting research studies on generating pH values at the surface ocean by providing global maps of seawater pH for depth levels up to 2000 m, essential for understanding ocean acidification and its impacts on ecosystems in the ocean interior.

**General comments**:

Machine learning offers flexible frameworks to link in situ observations of marine carbonate system variables with relevant environmental conditions. The novelty of this research lies on the idea of mapping on direct pH data from the surface down to different depth layers instead of computing through the carbonate system speciation given prior parameters as pCO2, DIC, …. The authors are also able to integrate various data sources, including in-situ measurements and satellite-based datasets as input of pH estimates.

While the manuscript presents the effectiveness of machine learning models (SOM, Stepwise FFNN, FFNN), there is a crucial need for greater transparency regarding model architecture, selection process of predictors, hyperparameter tuning, model accuracy and uncertainty quantification. Including these details would enhance the reliability of the proposed product and interpretability of the study.

Response: Thanks for the suggestion. We have revised the methodology section for better clarity, the detailed changes are as the following responses.

**1.Methodologies:**

Even interpolation with direct pH data may be feasible but model accuracy remains limited due to the modest amount and data coverage of GLODAP pH used for training and validation, that may be much more problematic at the deep sea. More statistics on GLODAP data used for FFNN training and validation are needed for a comprehensive evaluation on the model efficiency. It would be worthy to test the proposed method by subsampling with the same GLODAP tracks on the exiting products that offer global maps of pH at some levels depths (see for instance: https://doi.org/10.48670/moi-00015).

Response: Thanks for the suggestion. As we only divided two vertical layer to train FFNNs, the

number of training samples at the deep sea is comparable with the mixed layer near surface, and only the depth coverage of these training samples was notably larger than the mixed layer. Although these training samples were sparser in space, the FFNNs suggested a low predicting pH at the deep sea, which may be caused by the smaller seasonal and annual pH variability at the deep sea.

We used about 75% of GLODAP data for training and about 25% for testing in each iteration of evaluation, with training and testing groups divided by years. After repeating four times and changing testing groups in each iteration, all GLODAP sample has been used for testing once to carry out a comprehensive evaluation. We have added a figure showing the statistics of training and testing sample in the supplement as the following Figure S1.

The mentioned product in https://doi.org/10.48670/moi-00015 starts from 2021, which does not coincide with the temporal coverage of our product and were not used for evaluation. Furthermore, other existing machine learning-based products are only available for the global surface ocean, so we only presented comparisons of surface pH trends with these products. For further evaluation of our product in deep sea, we have added comparison with qualified pH data from Biogeochemical Argo float dataset in the validation section.

[Figure]

Figure S1. Statistical distribution of GLODAP samples used for training and testing in each province. Iteration 1-4: repeated evaluation with different training and testing samples dividing by years. Samples in 1992, 1996, …, 2020 were used for testing and the rest were used for training in

iteration 1; samples in 1993, 1997, ..., 2017 were used for testing and the rest were used for training in iteration 2.

The use of all the three ML algorithms seems redundant and would gain uncertainty for pH mappings (particularly with limited number of training data). FFNN itself would be skillful enough to interpolate pH without SOM for regional clustering and stepwise FFNN for predictor selection (see for instance: Broullón et al., 2019; 2020; Chau et al., 2022; 2024). Results derived from SOM shown in this study (e.g. Figures 1 and 8) does do not reflect the sharing patterns of mechanisms in CO2 uptake and storage, e.g., at temperate (equatorial) zones between different oceanic basins. pH estimation based on clustered biomes creates discontinuity at the regional boundary. Clustering and predictor selection are probably available in FFNN training phase as it automatically weights and select input neurons to get optimal output. In addition, the use of lat, lon, depth as predictors potentially locallizes training data and would allow to interpolate pH at non-observed locations. Results after Stepwise FFNN are not accurate enough (see comment for Table 2 in Specific Comments).

Response: Thanks for the suggestion. SOM-based regional clustering has also been proved effective in reducing regional predicting error in machine learning mapping of carbonate system variables, such as Landschützer et al. (2016), Iida et al. (2021), and Zhong et al. (2022). Although there is discontinuity problem, we decided to use SOM clustering based on the following main reasons:

(1) It is difficult to adjust the FFNN architecture and input predictor to obtain the optimal performance in different regions and depths. The number of neurons and combination of predictors adjusted to reduce predicting error in specific region may also lead to higher errors in other regions. Using regional-specific predictors and model architectures can reduce errors in different regions simultaneously.

(2) Inputting over 300 thousand unbalanced GLODAP samples into one FFNN model may generate biased outputs in regions with sparser samples. Previous research suggested the one FFNN trained with unbalanced samples will generally output values biased toward the majority pattern of training samples (Zhong et al., 2024). The one FFNN model only get optimal output in most data-rich areas, such as north Pacific and north Atlantic with the data amount far more than other areas. The output pH value in data-sparse areas may be more biased toward pH pattern in data-rich areas.

(3) The discontinuity problem can be solved with the further accumulate of pH measurements and

improvement in SOM technical. In the surface ocean, the discontinuity problem did not appear in most boundaries.

(4) Training only one FFNN is also feasible, but is preferable in the method that mapping DIC and TA first and then calculating other variables, as these two variables are relatively more conservative.

Sample position and time were used as predictors because the collected environmental variables are not enough comprehensive to cover all ocean processes affecting pH, as gridded products of many variables are currently not available or only climatological. If more environmental variables are included in future works, the sample position and time will fail to compete with other environmental variables in the predictor selection procedure.

Iida, Y., Takatani, Y., Kojima, A., & Ishii, M. (2021). Global trends of ocean $CO_2$ sink and ocean acidification: an observation-based reconstruction of surface ocean inorganic carbon variables. *Journal of Oceanography*, *77*, 323-358.

Landschützer, P., Gruber, N., & Bakker, D. C. (2016). Decadal variations and trends of the global ocean carbon sink. *Global Biogeochemical Cycles*, *30*(10), 1396-1417.

Zhong, G., Li, X., Song, J., Qu, B., Wang, F., Wang, Y., ... & Duan, L. (2022). Reconstruction of global surface ocean p $CO_2$ using region-specific predictors based on a stepwise FFNN regression algorithm. *Biogeosciences*, *19*(3), 845-859.

Zhong, G., Li, X., Song, J., Wang, F., Qu, B., Wang, Y., ... & Dai, J. (2024). The Southern Ocean carbon sink has been overestimated in the past three decades. *Communications Earth & Environment*, *5*(1), 398.

Please consider to provide more details on the data preprocessing steps, including how different data sources were transformed and harmonized, as well as elaborate on (1) the choice of many predictors (e.g., why temperature anomaly was used instead of the standard ones; why both number of months and Year Month are needed,...), (2) machine learning algorithms used, and (3) the rationale behind selecting them over other potential models.

Response: The collected products from different sources are all gridded datasets and the most are in the same 1° resolution. In the preprocessing step, products with higher resolution were converted into 1° resolution by averaging all data within the same 1° grid into one value.

The predictors were selected from collected environmental variables and sample information, which are expect to be as many as possible and cover most ocean processes affecting

pH. Most variables have been used for reconstruction of ocean carbonate system variables in previous researches, such as temperature and its anomaly. Some variable seems redundant but have different features. For example, the sample time has no seasonal cycle pattern information when using the number of months and Year, and is disconnected between years when using Months as 1-12. Therefore, we collected as many variables as possible and then selected predictors using a Stepwise regression algorithm based on FFNNs (referred as Stepwise FFNN), according to the pH predicting errors when using different combination of variables as FFNN inputs. This algorithm has been proved to have capacity to identity the most informative variables in previous $pCO_2$ mapping research and can effectively reduce regional predicting errors. We have revised the section 2.3 pH product construction for better clarity of the predictor selection and product construction procedure.

**2.Model robustness and Uncertainty Analysis**:

Although the authors have shown the validation against independent datasets from GLODAP, other ML-based methods, and some time series of direct measurements (e.g., HOT, ESTOC, BATS), the examination limited almost for the surface ocean. Evaluation and analysis of model accuracy at seawater depths would provide a clearer understanding of the model performance. Some timeseries stations have offered pH data below the surface (e.g., HOT, ESTOC, BATS), other sources for data evaluation can be found in Sutton et al., (2019), Lange et al., (2024).

Response: Thanks for the suggestion. We have added the BAT, HOT, and DYFAMED time series data below the surface in the validation section.

[Figure]

**Figure 7. RMSE and pH difference between FFNN pH and time series observations at different depths.** a) BAT station at 31°50' N, 64°10' W based on data from 1992 to 2020; b) HOT station at 22° 45' N, 158° 00' W based on data from 1992 to 2020; c) DYFMED station at 42.3°N, 7.5° E based on data from 1998 to 2017.

Despite higher RMSE at certain depths, the RMSE at most depths in the deep areas of the BAT station and DYFAMED station is below 0.03, indicating that the notable deviations may

only occur at local scale. This notable deviation may be due to sparser GLODAP measurements in certain areas, or difference in depths between pH product and independent observations. For example, the FFNN pH product was reconstructed at depths of 1800 m and 2000 m in the bottom. If the time series observation is at 1910 m depth, it will be compared with the FFNN pH value at 2000 m in the independent validation. This depth difference significantly increases the pH error in validation based on independent data if the number of independent observations was limited.

The manuscript would benefit from a more comprehensive uncertainty analysis. It is not convinced that adding the source of RMSE from FFNN [H+] is essential for the quantification of pH uncertainty as the author ultimately produce 4D pH fields from GLODAP pH data (not [H+]. Presenting a comparison between RMSE of pH derived from direct GLODAP pH and from [H+] in the supplementary is more appropriate. Instead, it would be worthy to consider predictors' uncertainty in the total uncertainty of reconstructed pH.

Response: Thanks for the suggestion. Adding the source of RMSE from FFNN $[H^+]$ is mainly to distinguish the uncertainty between areas with same pH RMSE but different pH levels, as the uncertainty would be the same between these areas if calculated directly based on pH RMSE. The section of comparison between pH and [H+] has been moved to the supplementary. Including all predictors' uncertainty will be a better way to estimate the pH product uncertainty. However, as described in the supplementary section *Uncertainty and construction method of selected ocean products*, the uncertainty of particular predictor products is unclear. It is not feasible to convert the uncertainty of predictor products through the FFNN when some inputs are missing. Therefore, we have to estimate the pH product uncertainty in a different way.

**Specific Comments:**

Lines 44-45: The two studies report fast decrease in pH in the ocean interior. The authors should reword this sentences.

Response: The fast decrease in pH was reported in subsurface above 500 m in these two studies, but a relatively slow acidification can be also observed in deeper areas. The sentence has been corrected as the following:

"Meanwhile, relatively slow acidification was found in the deep Atlantic Ocean below 2000 m (Guallart et al., 2015), and rising pH in deep waters around 1000 m was also reported in the North Pacific Ocean (Ishizu et al., 2021)."

Lines 45-46: Can be rephrased (for instance: "there remains a need to enhance our understanding of global ocean acidification rates across varying depths.")

Response: Thanks for the suggestion. The sentence has been modified as the following:

"With limited reports about acidification below the surface, there remains a need to enhance our understanding of global ocean acidification rates across varying depths."

Line 49: "the global mapping of" to "global reconstructions of".

Response: The sentence has been corrected following the suggestion.

Lines 50-51: Gregor and Gruber, (2021) and Chau et al., (2024) have published full datasets of many carbonate variables (pH, DIC, Alkaninity included).

Response: The sentence has been modified as the following:

"Recent applications of machine learning methods in global reconstructions of marine carbonate system variables have facilitated global-scale research on the acidification and carbon cycle, including the single/ensemble-based FFNN method and the SOM-FFNN method for mapping surface ocean partial pressure of $CO_2$ ($p$$CO_2$, Landschützer et al., 2014; Chau et al., 2022; Zhong et al., 2022; Chau et al., 2024), dissolved inorganic carbon (DIC, Broullón et al., 2020; Keppler et al., 2020; Gregor and Gruber, 2021; Chau et al., 2024), and alkalinity (Broullón et al., 2019; Gregor and Gruber, 2021; Chau et al., 2024)."

Lines 55-56: "The construct pH product"; this term is not a standard scientific or technical term. Maybe replace with "the proposed product" or "the recontruced pH data".

Response: The term "The construct pH product" has been replaced by "the proposed pH product".

Line 61: (Lauvset et al., 2022) → update refs for GLODAPv2.2023 and cite right after 2023 version instead.

Response: The citation has been updated.

Line 62: "in-situ temperature"; Reviewer do not see any specific role of temperature in pH mappings throughout the manuscript.

Response: The in-situ temperature was mentioned here is only for indicating that this product is not at 25°C, as the GLODAP dataset also provides pH data corrected to 25°C.

Line 63: Table 1 show most of predictors' products with no depth levels, it is not clear how to map pH with constant values of predictors over depths!

Response: The product of temperature, salinity, nutrient concentration, dissolved oxygen, DIC, and alkalinity provide values across different depths.

Lines 66-70: many predictors have been used but there are no hints (citations) showing why they

should be included in model fitting.

Response: The citations of previous application of the predictors have been added.

Line 80: "such as the ocean currents product" → should be at the beginning of the sentence.

Response: This term has been moved to the beginning of the sentence.

Table 1: please mention temporal and vertical resolutions of predictors' products used; comments on how to derive transformation for Date (Year, Month,...).

Response: The temporal and vertical resolution have been added in Table 1. Product with daily or weekly resolutions were converted to the monthly resolutions by directly averaging all values within the same month, and there is no product with year resolution.

Line 116: "Therefor".

Response: The typo has been corrected.

Figure 1: consider to not use SOM (see in General comments).

Response: Using one FFNN model can indeed reconstruct global gridded pH data, but this method does not account for the regional differences in factors affecting pH. The primary feature of our method is the consideration of regional differences in pH drivers, allowing for the selection of the most suitable predictors for pH reconstruction in each region, thereby increasing accuracy. Despite continuity issues at some boundaries, the overall reconstruction error is lower than using one model. In our previous research on reconstructing gridded $pCO_2$ data, we also employed the same SOM method to use regional-specific predictors, which significantly reduced reconstruction errors and was well-received by peers.

Lines 153-154: "To mitigate the influence of the FFNN's initial state on predicting values, multiple networks with the same structure but different initial states were trained and their results were averaged"; standard deviation from output averaging should be reported.

Response: The figure showing mean standard deviation between FFNN pH with different initial status has been added in the supplement as the following:

[Figure]

**Figure S5. Mean standard deviation between FFNN pH with different initial status.**

Tables 2, 3: consider to not use Stepwise FFNN to select predictors. FFNN itself would choose which inputs benefit model training. Results from Stepwise FFNN are not align with marine CO2 system features and driving mechanisms. For instance, temperature is one of the key factors modulating CO2 absorption over the Arctic and subpolar regions (thus performs impact on pH); however, this predictor was not chosen after Stepwise FFNN. In addition, please clarify the use of both temperature (salinity,...) and their anomalies that are redundant information and may challenge FFNN training!!!

Response: In current commonly used methods for reconstruction of global ocean gridded data, the predictors are typically selected empirically. The manually selected predictors vary in different methods for constructing the same marine chemical variable, contributing to partial differences between the products. Our algorithm selects input predictors based on statistical characteristics, eliminating the influence of subjectivity and randomness of manual selection. In some regions, effective predictors for reducing pH reconstruction errors may differ from predictors identified based on experience. For example, in the Arctic Ocean, temperature is not used because its effect on pH is already reflected in other input predictors like $p$CO$_2$, which is

notably affected by temperature. The $p$CO$_2$ increases with the increasing temperature. Additionally, the seasonal temperature variation in the Arctic Ocean is small, so other parameters can sufficiently reflect the effect of temperature. Including temperature as a pH predictor for the Arctic Ocean increased the pH reconstruction error estimated based on GLODAP samples, which is why we excluded temperature when reconstructing pH for the Arctic Ocean.

The monthly anomalies of temperature only reflect seasonal and interannual variations, removing the regional distribution features of temperature. In the 50-60S region of the Southern Ocean, directly using temperature as input predictors primarily reflects the spatial distribution characteristics of temperature, as the scale of regional differences in temperature are much greater than seasonal and interannual variations. This disturbed the model's learning of seasonal and interannual temperature changes. Using both temperature and its monthly anomalies to reflect regional distribution and temporal changes can reduce pH reconstruction errors. Therefore, both temperature and its monthly anomalies are used as pH predictors in the subpolar Southern Ocean. Similarly, other variables are generally not used together with their monthly anomalies in most regions. They are only used together when it can reduce pH reconstruction errors.

Line 174: RMSE; the metric for validation is not consistent with training (MAE, Line 135).

Response: This is because the extreme values have higher weighting if the reconstruction error was represented by RMSE compared to the MAE. In the selection procedure, the algorithm was design to focus on reducing errors for the majority of testing samples rather than particular samples with extreme values. The MAE can also be used to present FFNN performance in the validation section, but RMSE is more commonly used. So we used RMSE in the validation section.

Line 183: Please clarify how to define H+ here.

Response: [H$^+$] is the molar hydrogen ion concentration here.

Lines 191-206: It's worthy to rework on this section: It is not ease to interpret the uncertainty quantified with pH0 and sigma (see also in General Comments for details).

Response: Although this method is not commonly used, it can better represent the uncertainty of the reconstructed pH products. This method includes the main factors influencing uncertainty, including the pH reconstruction errors and the notable differences in [H$^+$] that caused by the same

pH error at different pH levels. For example, the same pH error of 0.02 lead to a difference in $1.42*10^{-9}$ M of $[H^+]$ when pH is 7.5, and lead to a difference in $1.42*10^{-10}$ M of $[H^+]$ when pH is 8.5. The difference in $[H^+]$ differs notably but the uncertainty directly estimated by pH errors will be the same.

Lines 212-214: "A better performance of the FFNN was found in the intermediate layer, with testing samples more concentrated on the y=x line. The RMSE in the mixed layer is 0.034, higher than 0.026 in the intermediate layer." Reviewer would expect to see reverse results: there exist very few pH data and predictors' information which are able to support pH estimation in the deep sea than the shallower layer !!! Reconstruction errors (Uncertainty) would much higher there than the surface.

Response: Although the pH measurements are much sparser in the deep sea, we trained the FFNN using all samples in the deep sea from mixed layer depth to 2000 m. The number of training samples in the deep sea is even more than that in the mixed layer, as covered depths are much broader. Additionally, the FFNN underestimation of seasonal amplitude and short-term fluctuations is the main source of reconstruction errors, which are notably smaller in the deep sea. This can be observed in the validation based on time series station, the reconstruction error is notably higher when pH seasonally peaked and troughed. The same pattern of decreasing RMSE with depth can also be observed in the 3D reconstruction of DIC and TA in previous research, such as Broullón et al., 2019 and Broullón et al., 2020.

Broullón, D., Pérez, F. F., Velo, A., Hoppema, M., Olsen, A., Takahashi, T., ... & van Heuven, S. M. A global monthly climatology of total alkalinity: a neural network approach. Earth System Science Data, 11, 1109-1127, https://doi.org/10.5194/essd-11-1109-2019, 2019.

Broullón, D., Pérez, F. F., Velo, A., Hoppema, M., Olsen, A., Takahashi, T., ... & Kozyr, A. A global monthly climatology of oceanic total dissolved inorganic carbon: a neural network approach. Earth System Science Data, 12, 1725-1743, https://doi.org/10.5194/essd-12-1725-2020, 2020.

Line 219: remove "predicting" here and elsewhere, FFNN pH is informative enough.

Response: The term "predicting" has been removed in all contexts.

Lines 224-225: Any clarifications to have errors lower in the deep sea than the surface. I would appreciate of any clarification.

Response: In the upper ocean above 2000 m, the underestimation of seasonal amplitude and

short-term fluctuations has greater impacts on reconstruction errors than that of sparse training samples in the deep sea. Therefore, the validation based on the GLODAP dataset show a decreasing RMSE with increasing depths.

Line 230: "predicting error" is not correct. Please use "prediction error" or "reconstruction error" instead.

Response: Thanks for the suggestion. This term has been replaced by "reconstruction error".

Lines 237-238: "The RMSE in the early years was relatively higher than in recent years, while the number of GLODAP measurements increased with the years (Figure 5c)". Adding curves for number of GLODAP pH in each subplot will help to evidence the statements for Fig 5.

Response: The number of GLODAP pH has been added in Figure 5c.

Lines 249-: Why do the results show for Stepwise FFNN while final reconstruction is done with FFNN?

Response: The reconstruction in this work was based on a two-step method that including the predictor selection by stepwise regression using FFNN and the FFNN fitting of non-linear relationship. So, we named the pH product as the Stepwise FFNN product to summarize the whole method.

Lines 251 - 254:

"The surface seawater pH of our Stepwise FFNN product decreased by $0.0017 \pm 0.0007$ yr-1 on average during the past three decades at the BAT station, close to the $-0.0018 \pm 0.0001$ yr-1 of BAT time series observations in the same period (Bates et al., 2020). At the ESTOC station, the Stepwise FFNNN product and time series observations were also well consistent, with the RMSE of only 0.009 and a similar long-term trend (Chau et al., 2022)." these quotations are not correctly mentioned in refs (Bates et al., 2020; Chau et al., 2022). The pH decreasing rate was about $0.0019 \pm 0.0001$ per yr over the period 1983-2020 in the former study. Please clarify that the authors have used the data to compute the trends by themselves. Furthermore, Chau et al., (2022) do not provide long-term trend estimates of pH but Chau et al., (2024).

Response: The citation has been corrected to González-Dávila et al. (2010) for ESTOC and Chau et al., (2024) for the CMEMS pH product. The pH trends from previous products in Table 4 were computed using data from 1992 to 2020 in the BAT and HOT station, and were computed using data from 1995 to 2010 in the ESTOC station, to eliminate the influence of different temporal coverage. The description of compute period has been added in remarks of Table 4.

Line 255: remove "only" in this paragraph and elsewhere as 0.01 in pH is indeed large

correspondingly to a difference in 26% of $H^+$ (acidity level)".

Response: Thanks for the suggestion. The word "only" has been removed.

Table 4: report uncertainty estimates for all other products.

Response: The uncertainty of other products has been added.

**References**

Broullón, D., Pérez, F. F., Velo, A., Hoppema, M., Olsen, A., Takahashi, T., Key, R. M., Tanhua, T., González-Dávila, M., Jeansson, E., Kozyr, A., and van Heuven, S. M. A. C.: A global monthly climatology of total alkalinity: a neural network approach, Earth Syst. Sci. Data, 11, 1109–1127, https://doi.org/10.5194/essd-111109-2019, 2019.

Broullón, D., Pérez, F. F., Velo, A., Hoppema, M., Olsen, A., Takahashi, T., Key, R. M., Tanhua, T., Santana-Casiano, J. M., and Kozyr, A.: A global monthly climatology of oceanic total dissolved inorganic carbon: a neural network approach, Earth Syst. Sci. Data, 12, 1725–1743, https://doi.org/10.5194/essd-12-1725-2020, 2020.

Chau, T. T. T., Gehlen, M., and Chevallier, F.: A seamless ensemble-based reconstruction of surface ocean pCO2 and airsea CO2 fluxes over the global coastal and open oceans, Biogeosciences, 19, 1087–1109, https://doi.org/10.5194/bg-19-10872022, 2022b.

Chau T T T, Gehlen M, Metzl N, Chevallier F (2024). CMEMS-LSCE: A global 0.25-degree, monthly reconstruction of the surface ocean carbonate system, Earth System Data, 16, 121–160, https://doi.org/10.5194/essd-16-121-2024.

Lange, N., Fiedler, B., Álvarez, M., Benoit-Cattin, A., Benway, H., Buttigieg, P. L., Coppola, L., Currie, K., Flecha, S., Gerlach, D. S., Honda, M., Huertas, I. E., Lauvset, S. K., Muller-Karger, F., Körtzinger, A., O'Brien, K. M., Ólafsdóttir, S. R., Pacheco, F. C., Rueda-Roa, D., Skjelvan, I., Wakita, M., White, A., and Tanhua, T.: Synthesis Product for Ocean Time Series (SPOTS) – a ship-based biogeochemical pilot, Earth Syst. Sci. Data, 16, 1901–1931, https://doi.org/10.5194/essd-16-1901-2024, 2024.

Sutton, A. J., Feely, R. A., Maenner-Jones, S., Musielwicz, S., Osborne, J., Dietrich, C., Monacci, N., Cross, J., Bott, R., Kozyr, A., Andersson, A. J., Bates, N. R., Cai, W.-J., Cronin, M. F., De Carlo, E. H., Hales, B., Howden, S. D., Lee, C. M., Manzello, D. P., McPhaden, M. J., Meléndez, M., Mickett, J. B., Newton, J. A., Noakes, S. E., Noh, J. H., Olafsdottir, S. R., Salisbury, J. E., Send, U., Trull, T. W., Vandemark, D. C., and Weller, R. A.: Autonomous seawater $p$CO2 and pH time series from 40 surface buoys and the emergence of anthropogenic

trends, Earth Syst. Sci. Data, 11, 421–439, https://doi.org/10.5194/essd11-421-2019, 2019.

---

## Referee Report (RR1)

**Review of: "A global monthly 3D-field of seawater pH over 3 decades: a machine learning approach" by G. Zhong et al.; submitted to Earth System Science Data**

First of all, I would like to thank the authors for the careful and thoughtful responses to my comments and suggestions. I believe their revisions have improved the manuscript, and the new details make it clearer to read. However, there are still some issues that deserve to be addressed before the manuscript can be published.

**Specific points to raise :**

- To avoid confusion regarding temperature, I suggest adding the following clarification: *"Here, we present a monthly four-dimensional 1°×1° gridded product of global seawater pH at total scale and in-situ temperature (without standardization to 25°C)."*

This will ensure that readers understand the product focuses on pH measurements, while avoiding any implication that it is also a temperature product. I recommend adding this clarification at line 60 and elsewhere when necessary.

- The use of sin(Lat) as a predictor is questionable since latitude is not circular.
Response: This normalization method was inspired from previous research, such as Denvil-Sommer, A., et al. (2019), where they also normalized latitude and longitude to radians using sine and cosine transformations. Also, we have corrected the description name in Table 1 to "Sine of (latitude · π/180°)", "Sine of (longitude · π/180°)", and "Cosine of (longitude · π/180°)". As we used the "sind" and "cosd" function (sind(latitude) equals sin(latitude · π/180°)) in MATLAB, the original description was misleading and has been corrected.

The use of sin(Lat) as a predictor remains questionable, as latitude is not a circular (or periodic) variable. The response mentions that this normalization method was inspired by previous research, such as Denvil-Sommer et al. (2019), and the correction to the description in Table 1 has been made. However, I still have concerns for the following reasons:

- Sine and cosine functions are typically applied to periodic variables, such as longitude or day of the year, where values "wrap around." Latitude, on the other hand, is not periodic—*lat = -90°* is not equivalent to *lat = 90°*.
- Furthermore, the expression *sin(lat · π/180°)* seems inappropriate, as radians conversion should account for the full range of latitude values. If anything, it would be *sin(lat · π/90°)*, but even this is not ideal for latitude.

Given these issues, I maintain that latitude is not a suitable candidate for inclusion via sine and cosine transformations.

- Clarify how depth is used as a predictor and whether it corresponds to the depth of retrieval of the output or if the FFNN estimates X values for X depth levels.
Response: Thanks for the suggestion. Depth was used in the same way as latitude or time-related variables in Table 1. The sample depths of GLODAP measurements were input into FFNNs during the

training process, and the depths of 41 depth layers defined as target output layers were input into FFNNs during the interpolation process to generate a product covering 0-2000m. The description has been added in the 2.1 section as the following: "Temporal and spatial sample information, including latitude, longitude, depth and sample time, was also used as supplementary variables. Latitude and longitude were normalized to radians using sine and cosine transformations, to present connected sample position information. The spatial sample position and time information of GLODAP measurements were input in the training of FFNNs, and the spatial position and time of defined 1° and monthly product grids were input into FFNNs during the interpolation process to output a gridded product."

Thank you for the explanation regarding how depth is used as a predictor in the FFNN models. However, I am still unclear as to why pressure (*pres*) is not systematically included as a predictor in every FFNN (as seen in Table 2). Given that depth-related information is a critical factor, especially in oceanographic models, it seems logical that *pres* would be consistently used alongside depth.

Additionally, this raises the broader question of why other key spatial-temporal predictors, such as longitude, latitude, and time, are not always systematically included as inputs in the FFNNs. It's unclear why time, in particular, is only integrated in some models and not others, given its fundamental importance in understanding temporal variability in the data.

I suggest providing a clearer rationale for the selective use of these variables and ensuring that key predictors are consistently applied across all FFNNs, or explaining why their inclusion is sometimes omitted.

- Adding a column to Table 1 to indicate which process each variable is associated with would be informative.
Response: Thanks for the suggestion. The related processes have been added in Table 1 as the following:
Thank you for incorporating the suggestion to add the related processes to Table 1. However, I believe further clarification is still needed for some variables. Specifically, variables like PAR, KD, RRS, and Ta/b may also be associated with the biological production of organic matter, as they are crucial in this context. Ensuring that these variables are linked to biological processes in the table will provide a more complete understanding of their roles. I don't think that 'Supplementary for lacking interannual variability of other variables, or potential correlation with unclear process affecting pH' is relevant for these variables.

-Line 191: The paragraph is unclear. The statement, "Therefore, the uncertainty of our pH product was directly estimated from the FFNN pH predicting errors, instead of synthesizing the inherent uncertainty of each used predictor product," needs further clarification. How was this done?
Response: As described in equation (2), the uncertainty was estimated from local pH value and pH predicting error in the corresponding province. For the uncertainty in certain grid, we first convert pH predicting error in the corresponding province into difference of [H+ ], by logarithm transfer of predicted and GLODAP measured pH and then calculating RMSE. Subsequently, the RMSE of [H+] was transferred to pH uncertainty based on the local pH value. $= -\log_{10}(10^{-pH_0} - RMSE_{[H^+]})$ $-pH$ where RMSE[H+] was the RMSE of [H+ ] converted from FFNN pH predicting error in each vertical layer and in each biogeochemical province. pH0 was the local predicted pH value in the grid

Thank you for the detailed response, but the explanation is still unclear regarding how the method described provides local uncertainties. Specifically, the statement *"the uncertainty of our pH product was directly estimated from the FFNN pH predicting errors, instead of synthesizing the inherent uncertainty of each used predictor product"* remains ambiguous.

While equation (2) describes converting pH prediction errors into RMSE for [H+] and then back to pH uncertainty, it's still not clear how this process provides uncertainty estimates at a local (grid-specific) level. Could you provide more detailed clarification on how this method operates for each grid and how the local predicted pH value ($pH_O$) factors into the uncertainty calculation? Additionally, a more intuitive explanation of why inherent uncertainty from each predictor was not feasible would help clarify this point for readers.

- Moreover, it would be interesting to add comparison against qualified pH data from BGC-Argo dataset.

Response: Thanks for the suggestion. Comparison of global scale trend has been added in Table 4. The BGC ARGO pH data qualified by IMOS has been added in the validation section. Different from the validation results based on the GLODAP dataset, the RMSE between FFNN pH and BGC ARGO pH data is higher in the deep ocean. Only the bias between FFNN pH and BGC ARGO pH data tends to increase with depth in most basins. In contrast, greater biases between FFNN pH and GLODAP pH occur mainly in the surface layer. Especially in the Southern Ocean, the bias between FFNN pH and GLODAP pH is nearly zero below 1000 m, notably lower than biases between FFNN pH and BGC ARGO pH data ranging from 0.053 to 0.076. This may be primarily attributed to the discrepancies between GLODAP dataset and the BGC ARGO dataset in the deep ocean, as our product was based on GLODAP dataset and small biases with GLODAP pH were observed in the deep ocean.

Thank you for incorporating the comparison with the BGC-Argo pH data into the validation section. However, there are several important points that still need to be addressed:

- *BGC-Argo* should be written with proper formatting (i.e., "BGC-Argo," not all caps for "Argo").
- It would be valuable to include a reference to the BGC-Argo dataset, such as Claustre et al. (2020) https://www.annualreviews.org/content/journals/10.1146/annurev-marine-010419-010956.
- As it is mentioned that only data qualified by IMOS were used, does this imply that the validation was limited to the Southern Ocean and data from CSIRO? If so, the validation is not truly global. For a more comprehensive validation, I suggest using data from all DACs (Data Assembly Centers), accessible via the GDACs (Global Data Assembly Centers). There are two GDACs available: US GDAC and France Coriolis GDAC (https://argo.ucsd.edu/data/data-from-gdacs/).
- A geographical map of BGC-Argo pH profiles should be included to visualize where the validation was performed.
- Specific details regarding the data used in the validation should be provided. Only delayed-mode pH-adjusted data with QC (Quality Control) 1 applied should be used for a robust comparison.

- Additionally, BGC-Argo should be acknowledged in the acknowledgments, following the guidelines here: https://argo.ucsd.edu/data/acknowledging-argo/.

---

## Referee Report (RR2)

**Review of: "A global monthly 3D-field of seawater pH over 3 decades: a machine learning approach" by G. Zhong et al.; submitted to Earth System Science Data**

First of all, I would like to thank the authors for the careful and thoughtful responses to my comments and suggestions. I believe their revisions have significantly improved the manuscript, and the new details make it clearer to understand.

There are two minor comments that I would like the authors to address before publication.

**Specific points to raise :**

1. **Depth as a predictor:**

   I apologize for the confusion caused by my earlier comments where I used "pressure" instead of "depth.". I fully understand and agree with the authors' rationale for excluding pressure due to its high correlation with depth. However, my concern pertains to the use of depth as an input predictor, which is not applied consistently across all bioregions. From the authors' first response, I understand that depth is used as input predictor to estimate pH at specific levels (e.g., one of the 41 defined depth levels). However, I remain unclear how pH at different depths is estimated in certain bioregions where depth is not included as an input (particularly in the mixed layer).

   For example, in the Subpolar North Atlantic bioregion, pH in the mixed layer is estimated using predictors such as Phosphate, DO, Nmon, DIC, Sal, and Bathy, but depth is not explicitly included. None of these environmental predictors can fully substitute for depth. This issue also applies to the Equatorial Atlantic and Subtropical South Atlantic in the mixed layer. By contrast, for intermediate layers, this concern does not arise as depth is consistently included.

   The paragraph the authors added regarding longitude, latitude, and time being replaceable by other environmental variables is very useful and improves clarity in the text. However, this point does not address the specific issue of how pH can be accurately retrieved for different depths when depth is not used as an input predictor.

2. **Validation using BGC-Argo data:**
   Considering the spatial distribution of BGC-Argo data, which is concentrated mainly in the Southern Ocean, I think it would be valuable to include the number of points (or profiles) used to compute the biases presented in Table 5. This information would help clarify the representativeness of the validation results.

---

## Author Response (AR2)

(Gray: previous comments and responses; Black: current comments; Red: current responses)

**Reviewer#2**

First of all, I would like to thank the authors for the careful and thoughtful responses to my comments and suggestions. I believe their revisions have improved the manuscript, and the new details make it clearer to read. However, there are still some issues that deserve to be addressed before the manuscript can be published.

**Response:** Thanks for your suggestions, which help improve the quality of our manuscript. We have revised the manuscript according to the raised remaining issues. The detailed changes are as the following responses.

**Specific points to raise :**

To avoid confusion regarding temperature, I suggest adding the following clarification: *"Here, we present a monthly four-dimensional 1°×1° gridded product of global seawater pH at total scale and in-situ temperature (without standardization to 25°C)."*

This will ensure that readers understand the product focuses on pH measurements, while avoiding any implication that it is also a temperature product. I recommend adding this clarification at line 60 and elsewhere when necessary.

**Response:** Thanks for the suggestion. We have added "(without standardization to 25°C)" following the description of in-situ temperature in the abstract and method section.

- The use of sin(Lat) as a predictor is questionable since latitude is not circular.

Response: This normalization method was inspired from previous research, such as Denvil-Sommer, A., et al. (2019), where they also normalized latitude and longitude to radians using sine and cosine transformations. Also, we have corrected the description name in Table 1 to "Sine of (latitude · π/180°)", "Sine of (longitude · π/180°)", and "Cosine of (longitude · π/180°)". As we used the "sind" and "cosd" function (sind(latitude) equals sin(latitude · π/180°)) in MATLAB, the original description was misleading and has been corrected.

The use of sin(Lat) as a predictor remains questionable, as latitude is not a circular (or periodic) variable. The response mentions that this normalization method was inspired by previous research, such as Denvil-Sommer et al. (2019), and the correction to the description in Table 1 has been made. However, I still have concerns for the following reasons:

• Sine and cosine functions are typically applied to periodic variables, such as longitude or day of the year, where values "wrap around." Latitude, on the other hand, is not periodic—*lat = -90°* is not equivalent to *lat = 90°*.

• Furthermore, the expression *sin(lat · π/180°)* seems inappropriate, as radians conversion should account for the full range of latitude values. If anything, it would be *sin(lat · π/90°)*, but even this is not ideal for latitude.

Given these issues, I maintain that latitude is not a suitable candidate for inclusion via sine and cosine transformations.

**Response:** Using the sine function is not for connecting the poles but rather normalizes the latitude to the

range of [-1, 1] to be consistent with the scale of transferred longitude. Therefore, we use sin(lat · π/180°) instead of sin(lat · π/90°), so that after transformation, lat = -90° becomes -1 and lat = 90° becomes +1, which are not the same. The reason for using the sine function is that the relationship between latitude and many natural processes, such as solar radiation and temperature, is typically nonlinear. The NOAA Greenhouse Gas Marine Boundary Layer Reference $xCO_2$ product used as a predictor was also based on sin(Lat) (Lan et al., 2023). Using the sine function better captures the nonlinear variation of pH and its influencing factors with latitude. Furthermore, using sin(lat) also incorporates additional information about the variation of grid area with latitude. The area of the 1° grid in high-latitude regions is significantly smaller than that in low-latitude regions, which is not indicated by latitude. When using sin(Lat), the difference in sin(Lat) between grid points is linearly related to the grid cell area, providing a better representation of the spatial information of used GLODAP pH observation data. Therefore, we used the sine function to standardize the latitude.

Lan, X., Tans, P., Thoning, K., & NOAA Global Monitoring Laboratory. NOAA Greenhouse Gas Marine Boundary Layer Reference - $CO_2$. [Data set]. NOAA GML, https://doi.org/10.15138/DVNP-F961, 2023.

- Clarify how depth is used as a predictor and whether it corresponds to the depth of retrieval of the output or if the FFNN estimates X values for X depth levels.

Response: Thanks for the suggestion. Depth was used in the same way as latitude or time-related variables in Table 1. The sample depths of GLODAP measurements were input into FFNNs during the training process, and the depths of 41 depth layers defined as target output layers were input into FFNNs during the interpolation process to generate a product covering 0-2000m. The description has been added in the 2.1 section as the following: "Temporal and spatial sample information, including latitude, longitude, depth and sample time, was also used as supplementary variables. Latitude and longitude were normalized to radians using sine and cosine transformations, to present connected sample position information. The spatial sample position and time information of GLODAP measurements were input in the training of FFNNs, and the spatial position and time of defined 1° and monthly product grids were input into FFNNs during the interpolation process to output a gridded product."

Thank you for the explanation regarding how depth is used as a predictor in the FFNN models. However, I am still unclear as to why pressure (*pres*) is not systematically included as a predictor in every FFNN (as seen in Table 2). Given that depth-related information is a critical factor, especially in oceanographic models, it seems logical that *pres* would be consistently used alongside depth.

Additionally, this raises the broader question of why other key spatial-temporal predictors, such as longitude, latitude, and time, are not always systematically included as inputs in the FFNNs. It's unclear why time, in particular, is only integrated in some models and not others, given its fundamental importance in understanding temporal variability in the data.

I suggest providing a clearer rationale for the selective use of these variables and ensuring that key predictors are consistently applied across all FFNNs, or explaining why their inclusion is sometimes omitted.

**Response:** Thanks for the suggestion. The reason for not using pressure is the high correlation between

pressure and depth. Different from the fundamentals of physical or biogeochemical ocean models, the FFNN method is based on a non-linear relationship regression between output and input environmental variables. For the FFNN method, more input variables do not always mean better, which is also the reason for performing a predictor selection procedure before the reconstruction in this work. The input predictors with high co-correlation tend to cause FFNN overfitting and worse performance. For example, in previous research reconstructing the vertical profile of seawater alkalinity, the pressure and depth are also not used together (Broullón et al., 2019). Therefore, to avoid overfitting we did not use pressure alongside depth.

In certain regions, latitude, longitude, and time are not included as predictive parameters because they do not provide sufficient information or effectively reduce the FFNN pH predicting error. While latitude, longitude, and time are associated with variations in pH, they do not directly influence the spatial or temporal distribution pattern of seawater pH. Instead, these variables are only related to regular zonal distribution patterns or temporal trends in seawater pH driven by physical, chemical, and biological processes. For example, using time as a predictor offers the same information for all grids, with insufficient information to capture the regional differences in pH change rates. The key predictors are thus environmental variables related to the physical, chemical, and biological processes that directly affect pH. In some regions where the environmental variables sufficiently reflect the factors influencing pH or where spatial and temporal pH patterns are not notable, adding latitude, longitude, and time as predictors does not contribute sufficient information and can not effectively reduce predicting errors. Instead, it may lead to the overfitting of FFNN, overlooking pH fluctuations within small regions or over short time scales. Therefore, latitude, longitude, and time were excluded from predictors in these areas.

The description of the selective use of sampling location and time information as predictors was added in section as the following :

"Spatial and temporal variables, such as latitude, longitude, and time, are directly related to the spatial or temporal pH patterns rather than the factor driving pH variations. This means these variables are often co-correlated with other input environmental variables. In some regions where the environmental variables sufficiently reflect the factors influencing pH or where spatial and temporal pH patterns are not notable, adding latitude, longitude, and time as predictors does not contribute sufficient information and cannot effectively reduce predicting errors due to the co-correlation with other predictors. In this case, these spatial-temporal variables are not selected as predictors  (Tables 2 and 3). "

Broullón, D., et al. (2019). A global monthly climatology of total alkalinity: a neural network approach. Earth System Science Data, **11**(3): 1109-1127.

- Adding a column to Table 1 to indicate which process each variable is associated with would be informative.
Response: Thanks for the suggestion. The related processes have been added in Table 1 as the following:
Thank you for incorporating the suggestion to add the related processes to Table 1. However, I believe further clarification is still needed for some variables. Specifically, variables like PAR, KD, RRS, and Ta/b may also be associated with the biological production of organic matter, as they are crucial in this context. Ensuring that these variables are linked to biological processes in the table will provide a more complete understanding of their roles. I don't think that 'Supplementary for lacking interannual variability of other variables, or potential correlation with unclear process affecting pH' is relevant for

these variables.

Response: Thanks for the suggestion. We have modified the description of the linked processes of used products in Table 1 as the following:

PAR and KD490 are related to the light penetration and availability in aquatic systems influencing phytoplankton photosynthesis;

RRS and Ta/b are related to the phytoplankton composition and suspended particulate matter as indicators of biological productivity.

-Line 191: The paragraph is unclear. The statement, "Therefore, the uncertainty of our pH product was directly estimated from the FFNN pH predicting errors, instead of synthesizing the inherent uncertainty of each used predictor product," needs further clarification. How was this done?

Response: As described in equation (2), the uncertainty was estimated from local pH value and pH predicting error in the corresponding province. For the uncertainty in certain grid, we first convert pH predicting error in the corresponding province into difference of $[H^+]$, by logarithm transfer of predicted and GLODAP measured pH and then calculating RMSE. Subsequently, the RMSE of $[H^+]$ was transferred to pH uncertainty based on the local pH value.

$$\sigma = -\log_{10}(10^{-pH_0} - RMSE_{[H^+]}) - pH_0$$

where RMSE[H+] was the RMSE of $[H^+]$ converted from FFNN pH predicting error in each vertical layer and in each biogeochemical province. $pH_0$ was the local predicted pH value in the grid that uncertainty was estimated. Due to missing inherent uncertainty of particular predictor product, estimating uncertainty from inherent uncertainty of used predictor products was unfeasible.

Thank you for the detailed response, but the explanation is still unclear regarding how the method described provides local uncertainties. Specifically, the statement *"the uncertainty of our pH product was directly estimated from the FFNN pH predicting errors, instead of synthesizing the inherent uncertainty of each used predictor product"* remains ambiguous. While equation (2) describes converting pH prediction errors into RMSE for $[H^+]$ and then back to pH uncertainty, it's still not clear how this process provides uncertainty estimates at a local (grid-specific) level. Could you provide more detailed clarification on how this method operates for each grid and how the local predicted pH value ($pH_0$) factors into the uncertainty calculation? Additionally, a more intuitive explanation of why inherent uncertainty from each predictor was not feasible would help clarify this point for readers.

Response: Thanks for the suggestion. Equation (2) is calculated for all grids separately, yielding the local uncertainty for each grid. In the calculation of Equation (2), the $pH_0$ is the local predicted pH value, and the RMSE is the FFNN error of the province to which the grid belongs, which is used as the local error. The resulting σ represents the local uncertainty for the certain grid. In this approach, since not all grids have observational data to calculate the local error, the local predicted pH value serves to convert the overall province FFNN error into local errors for contained grids by logarithm transfer, allowing for the calculation of local uncertainty, and also play an important role in differentiating the uncertainty among grids with the same error but different pH values.

The reason why it is not feasible to calculate the total uncertainty by combining the inherent uncertainties of different predictor products is that the calculated via the error propagation relies on the partial derivatives of pH to each predictor. However, since the non-linear relationships established by the

FFNN do not have a specific formula, calculating the partial derivatives is extremely difficult. Therefore, it is not possible to use the basic error propagation method to integrate the inherent uncertainties of different products.

The original text has been modified as the following:

"Subsequently, the pH values were shown as pH$_0$±σ at each given pH0 value, and the local uncertainty σ stem from FFNN reconstruction errors was calculated as the following:

$$\sigma = -\log_{10}(10^{-p \ 0} - RMSE_{[H^+]}) - pH_0 \tag{2}$$

where RMSE$_{[H+]}$ was the RMSE of [H$^+$] converted from FFNN pH in each layer of all 14 biogeochemical provinces, pH$_0$ was the local FFNN predicted pH value. The local uncertainty σ calculated by this method is simultaneously related to the pH reconstruction error and local pH level which serves to convert the overall province FFNN error into local errors and better distinguishes the differences in uncertainty across different regions. The uncertainty of products used as pH predictors is one ineluctable source for pH reconstruction errors of the FFNN model. However, the direct estimation of pH uncertainty from summing the uncertainty of each used product is not feasible. Combining the inherent uncertainties of different predictor products via error propagation relies on the partial derivatives of pH to each predictor, but the non-linear relationships established by the FFNN do not have a specific formula, leading to the difficulty in calculating the partial derivatives. Therefore, the local uncertainty of our pH product was directly estimated from the regional FFNN pH reconstruction errors and local pH values following formula (2), instead of synthesizing the inherent uncertainty of each used predictor product through the propagation of errors."

- Moreover, it would be interesting to add comparison against qualified pH data from BGC-Argo dataset.

Response: Thanks for the suggestion. Comparison of global scale trend has been added in Table 4. The BGC ARGO pH data qualified by IMOS has been added in the validation section. Different from the validation results based on the GLODAP dataset, the RMSE between FFNN pH and BGC ARGO pH data is higher in the deep ocean. Only the bias between FFNN pH and BGC ARGO pH data tends to increase with depth in most basins. In contrast, greater biases between FFNN pH and GLODAP pH occur mainly in the surface layer. Especially in the Southern Ocean, the bias between FFNN pH and GLODAP pH is nearly zero below 1000 m, notably lower than biases between FFNN pH and BGC ARGO pH data ranging from 0.053 to 0.076. This may be primarily attributed to the discrepancies between GLODAP dataset and the BGC ARGO dataset in the deep ocean, as our product was based on GLODAP dataset and small biases with GLODAP pH were observed in the deep ocean.

Thank you for incorporating the comparison with the BGC-Argo pH data into the validation section. However, there are several important points that still need to be addressed:

• *BGC-Argo* should be written with proper formatting (i.e., "BGC-Argo," not all caps for "Argo").

Response: Thanks for the suggestion. The formatting has been corrected to "BGC-Argo" in all places.

• It would be valuable to include a reference to the BGC-Argo dataset, such as Claustre et al. (2020) https://www.annualreviews.org/content/journals/10.1146/annurev-marine-010419-010956.

Response: The reference to the BGC-Argo dataset has been added as the following:

Argo. Argo float data and metadata from Global Data Assembly Centre (Argo GDAC). SEANOE, https://doi.org/10.17882/42182, 2024.

Claustre, H., Johnson, K. S., & Takeshita, Y. Observing the global ocean with biogeochemical-Argo. Annual Review of Marine Science, 12(1), 23-48, https://doi.org/10.1146/annurev-marine-010419-010956, 2020.

- As it is mentioned that only data qualified by IMOS were used, does this imply that the validation was limited to the Southern Ocean and data from CSIRO? If so, the validation is not truly global. For a more comprehensive validation, I suggest using data from all DACs (Data Assembly Centers), accessible via the GDACs (Global Data Assembly Centers). There are two GDACs available: US GDAC and France Coriolis GDAC (https://argo.ucsd.edu/data/data-from-gdacs/).

Response: Thanks for the suggestion. The dataset of IMOS-Argo Profiles-biogeochemical data (https://catalogue-imos.aodn.org.au/geonetwork/srv/eng/catalog.search#/metadata/2223b7f2-4bac-4ff1-9b1e-aae9ac58deef) we used in previous revision also includes global BGC-Argo profiles. However, the used data contains both real-time mode and delayed-mode pH-adjusted data. According to the suggestion for robust validation, only delayed-mode data from France Coriolis GDAC has been used now (https://doi.org/10.17882/42182). However, much of the delayed-mode data concentrates on years after 2020, out of the range of our product. The used delayed-mode data is currently mainly distributed in the Southern Ocean (see Figure S6 in the next response). The revised section is as the following:

**3.1.2 Validation based on BGC-Argo float pH measurements**

Comparison with time series observations in deeper oceans suggested that the distribution of pH reconstruction errors with depth varies notably across different stations. To better assess the performance of FFNN in the reconstruction of pH at different depths, the FFNN reconstructed pH was further evaluated by comparing with independent BGC-Argo delayed-mode pH-adjusted data with quality control flag 1 at various depths (Argo, 2024), with spatial positions showing in Figure S6. Different from the validation results based on the GLODAP dataset, the RMSE between FFNN pH and BGC-Argo pH data in the intermediate layer is 0.051, higher than 0.035 in the mixed layer (Figures 8a and 8b). In both the mixed layer and intermediate layer, most samples were evenly distributed around the y=x line. However, in the intermediate layer, some samples were slightly offset and distributed below the y=x line, which may be the main reason for the notably higher RMSE between FFNN pH and BGC-Argo pH data in the intermediate layer. Overall, there is a good linear correlation between FFNN reconstructed pH and independent BGC-Argo pH data, with R² values of 0.73 and 0.84 in the mixed layer and intermediate layer, respectively.

[Figure]

**Figure 8. Difference between FFNN pH and BGC-Argo floats pH.** a) comparison between FFNN pH and BGC-Argo floats pH in the mixed layer; b) comparison between FFNN pH and BGC-Argo floats pH in the intermediate layer; c) Statistical

distribution of pH difference (FFNN pH minus BGC-Argo floats pH) at different depth levels. FFNN pH: pH data reconstructed in this work; BGC-Argo pH: pH data from BGC-Argo data from France Coriolis GDAC (Argo, 2024).

The distribution of pH differences between FFNN pH and BGC-Argo pH data at different depths reveals relatively smaller biases above 500 m (Figure 8c). However, below 500 m, the bias between FFNN pH and BGC-Argo pH data increases with depth and was the most remarkable at 2000 m. Comparing the pH bias calculated based on BGC-Argo dataset and GLODAP dataset, it is evident that only the bias between FFNN pH and BGC-Argo pH data tends to be more notable in deep areas except the Pacific Ocean (Table 5). In contrast, greater biases between FFNN pH and GLODAP pH occur mainly in the surface layer, with the most in the surface Indian Ocean. This disparity in distribution patterns between biases based on BGC-Argo dataset and GLODAP dataset is most remarkable in the Southern Ocean, where the bias between FFNN pH and GLODAP pH is nearly zero below 1000 m, compared to biases between FFNN pH and BGC-Argo pH data ranging from 0.040 to 0.068. These differences between FFNN pH and BGC-Argo pH data are primarily attributed to the discrepancies between GLODAP dataset and the BGC-Argo dataset in the deep ocean, as our product was based on the GLODAP dataset and small biases with GLODAP pH were observed in the deep ocean.

**Table 5. pH bias by area and depth computed with BGC-Argo and GLODAP dataset.**

| Area | | 0-50 m | 50-200 m | 200-500 m | 500-1000 m | 1000-1500 m | 1500-2000 m |
|---|---|---|---|---|---|---|---|
| Pacific | BGC-Argo | 0.028 | 0.016 | -0.003 | -0.013 | 0.027 | -0.004 |
| | GLODAP | -0.001 | -0.001 | 0.000 | 0.000 | 0.000 | -0.001 |
| Atlantic | BGC-Argo | 0.018 | 0.019 | 0.013 | -0.021 | 0.031 | 0.068 |
| | GLODAP | 0.000 | 0.000 | -0.001 | -0.001 | 0.000 | 0.000 |
| Indian | BGC-Argo | 0.023 | 0.034 | 0.025 | -0.022 | 0.000 | 0.036 |
| | GLODAP | -0.006 | -0.001 | -0.003 | -0.004 | -0.004 | -0.001 |
| Southern | BGC-Argo | 0.008 | 0.000 | 0.001 | 0.015 | 0.040 | 0.068 |
| | GLODAP | 0.004 | 0.001 | 0.001 | 0.000 | 0.000 | 0.000 |
| Global | BGC-Argo | 0.012 | 0.004 | 0.001 | 0.008 | 0.036 | 0.057 |
| | GLODAP | -0.001 | 0.000 | 0.000 | -0.001 | -0.001 | 0.000 |

• A geographical map of BGC-Argo pH profiles should be included to visualize where the validation was performed.

Response: The geographical map of BGC-Argo pH profiles has been added in the supplement as the following.

[Figure]

Figure S6. Station map of used delayed-mode BGC-Argo pH-adjusted data with quality control flag 1.

• Specific details regarding the data used in the validation should be provided. Only delayed-mode pH-adjusted data with QC (Quality Control) 1 applied should be used for a robust comparison.

Response: Thanks for the suggestion. The details about used BGC-Argo data were added as the following :

"For better evaluating the performance of FFNN below the surface, the constructed pH product was also compared to independent delayed-mode pH-adjusted data with quality control flag 1 from the biogeochemical-Argo (BGC-Argo) profiles from Global Data Assembly Centre (Claustre et al., 2020; Argo, 2024)."

• Additionally, BGC-Argo should be acknowledged in the acknowledgments, following the guidelines here: https://argo.ucsd.edu/data/acknowledging-argo/.

Response: The BGC-Argo has been acknowledged in the acknowledgments as the following:

"We thank BGC-Argo for sharing the pH float data. These data were collected and made freely available by the International Argo Program and the national programs that contribute to it (http://www.argo.ucsd.edu, http://argo.jcommops.org). The Argo Program is part of the Global Ocean Observing System. "

---

## Author Response (AR3)

**Reviewer # 1**

First of all, I would like to thank the authors for the careful and thoughtful responses to my comments and suggestions. I believe their revisions have significantly improved the manuscript, and the new details make it clearer to understand.

There are two minor comments that I would like the authors to address before publication.

Response: Thanks for your suggestions, which help improve the quality of our manuscript. We have revised the manuscript according to the comments. The detailed changes are as the following responses.

**Specific points to raise :**

1. **Depth as a predictor:**
   I apologize for the confusion caused by my earlier comments where I used "pressure" instead of "depth.". I fully understand and agree with the authors' rationale for excluding pressure due to its high correlation with depth. However, my concern pertains to the use of depth as an input predictor, which is not applied consistently across all bioregions. From the authors' first response, I understand that depth is used as input predictor to estimate pH at specific levels (e.g., one of the 41 defined depth levels). However, I remain unclear how pH at different depths is estimated in certain bioregions where depth is not included as an input (particularly in the mixed layer).

   For example, in the Subpolar North Atlantic bioregion, pH in the mixed layer is estimated using predictors such as Phosphate, DO, $N_{mon}$, DIC, Sal, and Bathy, but depth is not explicitly included. None of these environmental predictors can fully substitute for depth. This issue also applies to the Equatorial Atlantic and Subtropical South Atlantic in the mixed layer. By contrast, for intermediate layers, this concern does not arise as depth is consistently included.

   The paragraph the authors added regarding longitude, latitude, and time being replaceable by other environmental variables is very useful and improves clarity in the text. However, this point does not address the specific issue of how pH can be accurately retrieved for different depths when depth is not used as an input predictor.

   Reponse: In bioregions depth was not used as a predictor directly, pH at different depths was reconstructed based on input 3D field products of environmental variables used as predictors (e.g., nutrients, DO, TA, and DIC), which contain different values at the 41 defined depth levels. The vertical variability of these variables provided important information on how physical or biological conditions varied with depth. On this basis, the FFNN model learned how pH varied with depth from the vertical pattern of environmental variables used as predictors, which is notably similar to vertical pH pattern in bioregions depth was not used. The gridded seawater pH value at different depths was then accurately retrieved from the FFNN model and 3D field values of environmental variables used as predictors.

   For example, in the Irminger Sea station in the Subpolar North Atlantic bioregion, the phosphate concentration has a notably similar vertical pattern with seawater pH during different seasons (as shown in the following figure). Using phosphate and other 3D field variables (DO, DIC, and Sal) as predictors provided sufficient vertical distribution information in different seasons to reconstruct the vertical profile of seawater pH. Similarly, in the Equatorial Atlantic and Subtropical South Atlantic in the mixed layer, the FFNN model can also learn how pH varied with depths from the 3D fields of nitrate, silicate, and other environmental variables. However, the consistent vertical pattern only exists in specific regions. In the intermediate layers across wild depth ranges, more 3D fields of environmental variables and using depth

directly were necessary to provide sufficient vertical distribution information of seawater pH. Therefore, only in certain bioregions in the mixed layers was not used as a predictor, and depth was consistently included in the intermediate layers.

[Figure]

For better clarity, the description about how pH can be accurately retrieved for different depths has been added in the section as the following:

"In addition, depth is important in reconstructing the vertical pH distribution. However, it was not used as a predictor in certain regions of the mixed layer due to the notable similarity between the vertical pattern of pH and particular environmental variables used as predictors, such as phosphate, nitrate, and silicate. In this case, the FFNN model learned how pH varied with depth based on the similarity of vertical pattern between seawater pH and specific physical or biological conditions indicated by input environmental variables, and subsequently reconstructed seawater pH values at different depths using 3D fields of these environmental variables."

2. **Validation using BGC-Argo data:**
Considering the spatial distribution of BGC-Argo data, which is concentrated mainly in the Southern Ocean, I think it would be valuable to include the number of points (or profiles) used to compute the biases presented in Table 5. This information would help clarify the representativeness of the validation results.

Reponse: Thanks for the suggestion. We have added the number of samples used to compute the biases in Table 5 as the following:

**Table 5. pH bias by area and depth computed with BGC-Argo and GLODAP dataset.**

|  | Area |  | 0-50 m | 50-200 m | 200-500 m | 500-1000 m | 1000-1500 m | 1500-2000 m |
|---|---|---|---|---|---|---|---|---|
| Pacific | BGC-Argo | bias | 0.028 | 0.016 | -0.003 | -0.013 | 0.027 | -0.004 |
|  |  | N* | 16433 | 34708 | 36431 | 19840 | 8772 | 3565 |
|  | GLODAP | bias | -0.001 | -0.001 | 0.000 | 0.000 | 0.000 | -0.001 |
|  |  | N | 18687 | 26629 | 22746 | 24843 | 12613 | 13817 |
| Atlantic | BGC-Argo | bias | 0.018 | 0.019 | 0.013 | -0.021 | 0.031 | 0.068 |
|  |  | N | 3285 | 6832 | 7152 | 3565 | 1622 | 1288 |
|  | GLODAP | bias | 0.000 | 0.000 | -0.001 | -0.001 | 0.000 | 0.000 |
|  |  | N | 11808 | 15894 | 14330 | 18056 | 10686 | 11780 |
| Indian | BGC-Argo | bias | 0.023 | 0.034 | 0.025 | -0.022 | 0.000 | 0.036 |
|  |  | N | 407 | 916 | 920 | 491 | 241 | 57 |
|  | GLODAP | bias | -0.006 | -0.001 | -0.003 | -0.004 | -0.004 | -0.001 |
|  |  | N | 3145 | 5397 | 5124 | 5276 | 3457 | 3421 |
| Southern | BGC-Argo | bias | 0.008 | 0.000 | 0.001 | 0.015 | 0.040 | 0.068 |

| | | | | | | | | |
|---|---|---|---|---|---|---|---|---|
| Global | GLODAP | N | 66436 | 130563 | 135817 | 72564 | 27579 | 18692 |
| | | bias | 0.004 | 0.001 | 0.001 | 0.000 | 0.000 | 0.000 |
| | BGC-Argo | N | 7983 | 12268 | 10457 | 10341 | 6169 | 5800 |
| | | bias | 0.012 | 0.004 | 0.001 | 0.008 | 0.036 | 0.057 |
| | | N | 86561 | 173019 | 180320 | 96460 | 38214 | 23602 |
| | GLODAP | bias | -0.001 | 0.000 | 0.000 | -0.001 | -0.001 | 0.000 |
| | | N | 46415 | 66635 | 57491 | 62447 | 34994 | 37008 |

(*: N is the number of BGC-Argo or GLODAP samples used to compute the biases.)